



# The importance of cloud phase when assessing surface melting in an offline coupled firn model over Ross Ice shelf, West Antarctica

Nicolaj Hansen[1,2], Andrew Orr[3], Xun Zou[4], Fredrik Boberg[2], Thomas J. Bracegirdle[3], Ella Gilbert[3], Peter L. Langen[5], Matthew A. Lazzara[6,7], Ruth Mottram[2], Tony Phillips[3], Ruth Price[3], Sebastian B. Simonsen[1], and Stuart Webster[8]

[1]Geodesy and Earth Observation, DTU-Space, Technical University of Denmark, Lyngby, Denmark
[2]Danish Meteorological Institute, Copenhagen, Denmark
[3]British Antarctic Survey, High Cross, Madingley Road, Cambridge, UK
[4]Scripps Institute of Oceanography, La Jolla, USA
[5]iClimate, Department of Environmental Science, Aarhus University, Denmark
[6]Madison Area Technical College, Madison, Wisconsin
[7]University of Wisconsin–Madison, Madison, Wisconsin
[8]Met Office, Exeter, United Kingdom

**Correspondence:** Nicolaj Hansen (nih@dmi.dk)

**Abstract.** The Ross Ice Shelf, West Antarctica, experienced an extensive melt event in January 2016. We examine the representation of this event by the HIRHAM5 and MetUM high-resolution regional atmospheric models, as well as a sophisticated offline coupled firn model forced with their outputs. The model results are compared with satellite-based estimates of melt days. The firn model estimates of the number of melt days are in good agreement with the observations over the eastern and central

5    sectors of the ice shelf, while the HIRHAM5 and MetUM estimates based on their own surface schemes are considerably underestimated, possibly due to deficiencies in these schemes and an absence of spin-up. However, the firn model simulates sustained melting over the western sector of the ice shelf, in disagreement with the observations that show this region as being melt-free. This is attributed to deficiencies in the HIRHAM5 and MetUM output, and particularly a likely overestimation of nighttime net surface radiative flux. This occurs in response to an increase in nighttime downwelling longwave flux from

10    around 180-200 W m$^{-2}$ to 280 W m$^{-2}$ over the course of a few days, leading to an excessive amount of energy at the surface available for melt. Satellite-based observations show that this change coincides with a transition from clear-sky conditions to clouds containing both liquid-water and ice-water. The models capture the initial clear-sky conditions but seemingly struggle to correctly represent the ice-to-liquid mass partitioning associated with the cloudy conditions, which we suggest is responsible for the radiative flux errors.



## 1 Introduction

Intense and/or prolonged atmospheric-induced melting can result in widespread surface meltwater ponds over Antarctic ice shelves (Kingslake et al., 2017; Stokes et al., 2019). This can lead to the ice shelves thinning and even potentially collapsing if the meltwater enters the ice and results in enough pressure to cause hydrofracturing (Scambos et al., 2000, 2009; Kuipers Munneke et al., 2014), resulting in an increase in the discharge of grounded ice into the ocean and thus higher global sea levels (Dupont and Alley, 2005; Pritchard et al., 2012; Shepherd et al., 2018; Otosaka et al., 2023). Surface melting of ice shelves occurs when the upper surface temperature is greater than the freezing point of ice/snow of 0°C, as well as at sub-freezing temperatures (<0°C) if the snowpack consists of larger snow grains (Nicolas et al., 2017; Jakobs et al., 2021; Orr et al., 2023).

The relatively high temperatures that are associated with Antarctic ice shelf melting are usually in response to local and mesoscale circulations such as barrier winds, katabatic winds, and foehn winds (Orr et al., 2004, 2023; Coggins et al., 2014; Lenaerts et al., 2017a; Heinemann et al., 2019; Zou et al., 2021, 2023; Carter et al., 2022; Gilbert et al., 2022), as well as synoptic scale circulation patterns that facilitate the incursion of warm maritime airmasses, such as atmospheric rivers (Nicolas and Bromwich, 2011; Nicolas et al., 2017; Bozkurt et al., 2018; Scott et al., 2019; Wille et al., 2019, 2022; Turner et al., 2022; Orr et al., 2023; Zou et al., 2023). Therefore, to realistically capture local climate variability and simulate ice shelf melt patterns, it is essential to utilize regional atmospheric models at high spatial resolution, i.e., grid box sizes of the order 10 km or less. High-resolution simulations significantly enhance the description of crucial local-scale atmospheric processes and phenomena, particularly the complex forcing that characterises the Antarctic coastal margins, as well as resolving the smaller ice shelves that exist on spatial scales of 10-100 km (Hunt et al., 2004; Owinoh et al., 2005; Orr et al., 2005, 2014, 2023; Deb et al., 2018; Lenaerts et al., 2018).

An additional challenge faced by regional atmospheric models is to realistically represent the surface melting in response to atmospheric-induced warming and the resulting changes to the properties of snow/firn in the upper part of the ice shelf. This includes aspects such as meltwater production and ponding on the surface, snowmelt-albedo feedback, and retention and refreezing of liquid meltwater in the firn layer (Best et al., 2011; Trusel et al., 2015; Van Wessem et al., 2018; Walters et al., 2019; Jakobs et al., 2021; Keenan et al., 2021). The ability and sophistication of land surface and subsurface snow schemes in regional atmospheric models to represent these effects varies considerably, with the choice of spin-up time for the evolution of the snow/firn layer also being a factor in performance (Van Wessem et al., 2018; Carter et al., 2022). Dedicated and sophisticated offline coupled firn models serve as valuable tools to address these deficiencies (Langen et al., 2017; Keenan et al., 2021).

Cloud properties, particularly cloud phase and microphysics, are typically also challenging for regional atmospheric models to represent (Bodas-Salcedo et al., 2012; Abel et al., 2017; Hyder et al., 2018; Gilbert et al., 2020). For example, processes occurring at sub-grid scale, such as vapour deposition and turbulence, can influence the division of available water vapour between the solid and liquid phase, with consequent impacts on the radiative properties of the cloud (Furtado et al., 2016; Kim et al., 2020; Kretzschmar et al., 2020). Poor representation of these processes by the single-moment cloud microphysics



scheme used by the UK Met Office Unified Model (MetUM) has led to clouds containing too much ice water content and not enough liquid water content (Abel et al., 2017), leading to considerable biases in surface energy balance (SEB) and hence surface melting in Antarctica (King et al., 2015; Gilbert et al., 2020). For example, clouds with larger quantities of liquid water (relative to ice) are associated with higher downwelling longwave (LW) fluxes reaching the surface, while clouds containing more ice (relative to liquid) are associated with higher downwelling shortwave (SW) fluxes reaching the surface (Zhang et al.,
55 1996).

Properties such as cloud height, temperature and droplet/crystal size can also impact the radiative effect of the cloud, often in complex and contrasting ways (Lawson and Gettelman, 2014; Barrett et al., 2017; Gilbert et al., 2020). For example, errors with respect to the vertical distribution of liquid and ice, and especially the representation of thin supercooled liquid layers within mixed-phase clouds, can induce radiative biases (Gilbert et al., 2020; Vignon et al., 2021; Inoue et al., 2021). In addition to
microphysics, model cloud macrophysical parameterisations, especially relating to cloud fraction, may impact cloud radiative effects (Van Weverberg et al., 2023; McCusker et al., 2023).

Here we investigate the benefits of applying the sophisticated offline coupled firn model described by Langen et al. (2017) that represents key aspects such as the melt-albedo feedback to improve regional atmospheric model simulations of a prolonged and extensive episode of surface melt that occurred during January 2016 over the Ross Ice Shelf (RIS), West Antarctica. The
RIS frequently experiences major surface melt events due to both synoptic- and local-scale processes (Nicolas et al., 2017; Zou et al., 2021; Li et al., 2023; Orr et al., 2023), with this particular event attributed to an influx of warm and moist marine air, likely linked to a concurrent strong El Niño episode (Nicolas et al., 2017). The regional atmospheric model simulations examined were initially produced for Antarctic CORDEX (Antarctic COordinated Regional Downscaling EXperiment), and are based on HIRHAM version 5 (HIRHAM5) and MetUM version 11.1 (Orr et al., 2023). In these simulations, HIRHAM5 employed a
relatively sophisticated multi-layer snow scheme (Langen et al., 2015), while the MetUM utilized a simple composite snow/soil layer (Best et al., 2011).

Assessing the ability of models to estimate surface melt on Antarctic ice shelves is important for identifying deficiencies and aspects of the models that will require improvements in the future. Studies show that summertime surface melting of Antarctic ice shelves will likely increase considerably in the future (Trusel et al., 2015; Kittel et al., 2021; Feron et al., 2021; Gilbert and
Kittel, 2021; Boberg et al., 2022; van Wessem et al., 2023). For example, Trusel et al. (2015) suggests that scenario-independent doubling of Antarctic-wide melt will occur by 2050, and also that surface melt on several ice shelves under the high-emission climate scenario will approach the levels that contributed to the recent collapse of Larsen A and B ice shelves on the northern Antarctic Peninsula by 2100. Thus, improving the information on surface melting (and surface mass balance) and using this as an indicator for possible ice shelf collapse (Kuipers Munneke et al., 2014) or accelerations of outlet glaciers (Tuckett et al.,
2019) is vital for generating more accurate projections of future Antarctic ice sheet stability and its contribution to sea level rise (Fox-Kemper et al., 2021).



## 2  Methods and materials

The HIRHAM5 model combines the physics of the ECHAM5 general circulation model and the hydrostatic dynamical core of the HIRLAM7 numerical weather prediction model (Christensen et al., 2007). The model uses a single-moment microphysics scheme described by Sundqvist (1978). Furthermore, HIRHAM5 incorporates a five-layer snow scheme (extending to a depth of 10 m water equivalent) described by Langen et al. (2015), which calculates surface melt and the associated retention and refreezing of liquid water in the firn layer. The scheme also represents the dependence of snow albedo on temperature by linearly varying the albedo between 0.85 (for fresh dry snow/temperatures below -5°C) and 0.65 (for wet snow/temperatures at 0°C).

The MetUM version 11.1 model uses the Global Atmosphere 6.0 configuration (Walters et al., 2017, GA6), designed for grid scales of 10 km or coarser. This includes the ENDGame (Even Newer Dynamics for General atmospheric modelling of the environment) dynamical core, which solves equations for a non-hydrostatic, fully compressible, deep atmosphere. The model uses a single-moment cloud microphysics scheme based on Wilson and Ballard (1999). For simulating the thermal storage of snow it utilizes a "zero-layer" snow scheme described by Best et al. (2011), which employs a composite snow/soil layer and does not account for firn processes.

The physically-based multi-layer offline coupled firn model (hereafter referred to as the firn model) is based on the version implemented in HIRHAM5 (Langen et al., 2015), but heavily updated by Langen et al. (2017) to include 32 vertical layers (extending to a depth of 60 m water equivalent) and a sophisticated firn scheme. The model includes processes such as densification, snow grain growth, irreducible water saturation, impermeable ice layers, and snow state-dependent hydraulic conductivity. This enables a much more detailed representation of retention and refreezing of liquid water within the firn, and thus an improved representation of vertical water flow and refreezing. The version used here is identical to that previously applied to Antarctica by Hansen et al. (2021), which was based on the version optimized for Greenland (Langen et al., 2017; Mottram et al., 2017).

The HIRHAM5 and MetUM simulations were run over the standard Antarctic CORDEX domain (see Fig. 1) at a grid spacing of 0.11°(equivalent to 12 km) from 1979 to 2019, although in this study only output for January 2016 is examined. Lateral- and surface-boundary conditions for both simulations were provided by ERA-Interim reanalysis data (Dee et al., 2011). The HIRHAM5 simulation employed 31 vertical levels in the atmosphere (up to a height of 12.5 hPa), while the MetUM employed 70 vertical levels (up to a height of 80 km). Additionally, while the HIRHAM5 simulation uses a long-term continuous integration approach, the MetUM simulation uses a frequent re-initialisation approach (Lo et al., 2008). This consists of a series of twice-daily 24-hour forecasts (at 00 and 12 UTC), with output at T+12, T+15, T+18, and T+21 hrs from each of the forecasts concatenated together to form a seamless series of 3-hourly model outputs, with the output before T+12 hrs discarded as spin-up. The five-layer snow scheme used by HIRHAM5 was simply initialised and not spun-up. The zero-layer snow scheme used by the MetUM cannot be spun-up as it does not account for firn processes (Best et al., 2011). In any case, the frequent re-initialization approach used to produce the MetUM simulations would prevent the evolution of any internal snow/firn conditions. The setup for both models is identical to that described in Orr et al. (2023).



The firn model is subsequently driven by atmospheric forcing from the HIRHAM5 and MetUM simulations for the January 2016 period. This consists of 6-hourly averaged values of solid precipitation, liquid precipitation, surface evaporation, surface sublimation, surface downwelling SW radiative flux, surface downwelling LW radiative flux, sensible heat flux, and latent heat flux, which the firn model subsequently interpolates to hourly values before using them as forcing, by performing a linear

interpolation in time between the two nearest 6-hourly files. Prior to the simulations of the January 2016 period the firn model is spun-up for a period of 250 years using HIRHAM5 forcing (by repeating the same 1980s decade 25 times) to ensure a more realistic representation of the snow and firn properties. Following this, the firn model forced with MetUM output is spun-up for an additional 150 years using MetUM output (by repeating the same 1980s decade 15 times), to ensure that the firn pack has a memory of MetUM forcing.

The native surface melt output from the HIRHAM5 and MetUM simulations, as well as output from the HIRHAM5 and MetUM-forced firn model simulations, are used to calculate patterns of daily melt extent (defined as days with at least 3 mm of melt occurring) during the January 2016 event. For all snow/firn models, the energy flux used to melt the surface is calculated as the residual in the SEB whenever the surface temperature reaches above 0°C, after which it is reset to 0°C (Best et al., 2011; Langen et al., 2015, 2017). These are compared with daily melt extent estimates from satellite passive microwave

measurements at a grid spacing of 25 km (Picard et al., 2007; Nicolas et al., 2017), using the same melt threshold of 3 mm.

To better understand the physical processes responsible for the melt event, the characteristics and radiative properties of clouds are also examined, as well as their representation in HIRHAM5 and MetUM. Model estimates of the net surface radiative (SW + LW) fluxes, net surface LW fluxes, net surface SW fluxes, surface downwelling LW fluxes, and cloud liquid/ice water paths are compared with Cloud and the Earth's Radiant Energy System (CERES) satellite-based observations (Wielicki et al.,

1996; Loeb et al., 2018), which have a grid spacing of $1° \times 1°$. Vertical profiles of ice/liquid clouds were obtained from two orbits of the satellite-based Cloud-Aerosol Lidar and Infrared Pathfinder Satellite Observation (CALIPSO) instrument to further determine the phase of clouds during the event (Hu et al., 2009). Low-level cloud cover from the models are also compared with Moderate Resolution Imaging Spectroradiometer (MODIS) satellite-based imagery (Platnick et al., 2015). The MODIS cloud cover data produced at a grid spacing of $10 \times 10$ km are used here, as this agrees with the horizontal resolution of the

HIRHAM5 and MetUM models. However, as no direct observations were available at 12 UTC (over the western region of the RIS), a pseudo-image for this time was calculated by combining MODIS images that corresponded to satellite ground tracks passing over the western sector of the RIS at around 06 UTC (from the Aqua satellite) and 18 UTC (from the Terra satellite) and then averaging. Finally, model near-surface air temperatures are compared with measurements from four automatic weather stations (AWS) situated on the western sector of the RIS (Lazzara et al., 2012), referred to as Sabrina, Elaine, Schwerdtfeger,

and Marilyn. Figure 1 shows the locations of the four AWS and the two CALIPSO satellite ground tracks.



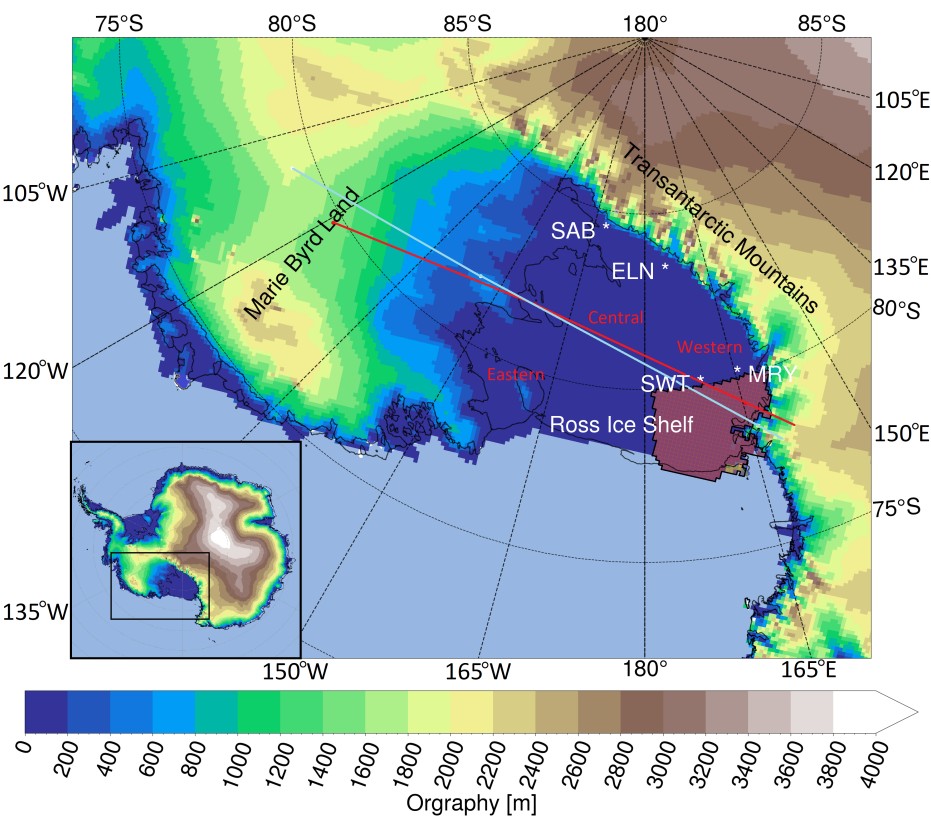

**Figure 1.** Map of West Antarctica showing the location of the Ross Ice Shelf, Marie Byrd Land, and the Transantarctic Mountains. Also labelled are the eastern, central, and western sectors of the ice shelf, with the eastern sector bordered by Marie Byrd Land and over the left-hand side of the ice shelf on the map, while the western sector is bordered by the Transantarctic Mountains and is over the right-hand side of the map. The orography (shading) and coastline (solid black line) are from the HIRHAM5 model. Also shown are the locations of four AWS situated on the western sector of the ice shelf; Sabrina, Elaine, Schwerdtfeger, and Marilyn, here referred to as SAB, ELN, SWT, and MRY respectively. The area over the western sector of the ice shelf that is investigated in depth is highlighted by the burgundy shading. The two solid lines crossing the ice shelf show the ground tracks of the CALIPSO satellite on the 14[th] (turquoise) and the 17[th] of January 2016. The inset map shows the full model domain used for the HIRHAM5 and MetUM simulations.

## 3 Surface melting

Figure 2 shows that extensive surface melting occurred over much of the central and eastern sectors of the RIS during January 2016, with the total number of satellite-observed melt days for this period approaching up to 15 in these locations. Examination of the observed melt pattern for individual days showed this period occurred roughly from the 11[th] to 25[th] of January (not shown). Much fewer melt days are observed over the western sector of the RIS during this period, with the transition between the high melt regime to the east and the low melt regime to the west abruptly occurring around 180°W, as also shown by Nicolas et al. (2017). Figure 2 also shows a considerable underestimation in the total number of melt days calculated from the


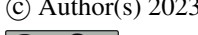



native melt output from HIRHAM5 and MetUM. This is especially apparent for HIRHAM5, which simulates only a few melt days over the eastern sector of the RIS and no melt days over the central sector. The MetUM performance is slightly better

in terms of both the number (up to 10) and pattern of the melt days, with the latter broadly agreeing with the observations. However, there is a considerable increase in the number of melt days calculated from melt output from the firn model forced by HIRHAM5 and MetUM, with up to 20 melt days simulated over the entire RIS. Although the firn model results are in better agreement with the observations over the eastern and central sectors of the RIS (albeit they now slightly overestimate the number of melt days in these areas), they erroneously simulate a much higher number of melt days (up to 20) over the western

sector of the RIS compared to the satellite-based observations.

To investigate further the discrepancies between the number of melt days estimated from the firn model and the observations, Fig. 3 compares model and satellite-based maps of daily melt area from the 13[th] to 18[th] of January. These six days were selected out of all of January because they (a) coincided with the main period of surface melting that occurred, and (b) showed the largest differences between the observations and firn model results, especially over the western RIS sector. The satellite-based

observations show a distinct melt-free region over the western sector of the RIS on each of these six days compared to the central and eastern sectors that show melt (i.e., broadly consistent with the observed surface melt pattern shown in Fig. 2a for all of January). This pattern is largely well simulated by the firn model from the 13[th] to 15[th] of January. However, from the 16[th] to 18[th] of January, the firn model results erroneously show a much smaller melt-free region over the western sector of the RIS compared to the observations, which gets progressively smaller each day. By the 17[th] and 18[th] of January any melt-free

area over the RIS is non-existent in the HIRHAM5-forced results and limited to the extreme western margins of the RIS in the MetUM-forced results.

The erroneous regions of melt over the western RIS simulated by the firn model during this period are consistent with near-surface air temperatures simulated by HIRHAM5 and MetUM being higher than the AWS-measured temperatures in this region, and especially at Schwerdtfeger and Marilyn (Fig. 4). In particular, at these two stations the HIRHAM5 near-surface

temperatures are consistently above -2°C from the 16[th] to 18[th], while the measurements are below this threshold. This is the threshold for melting for this particular event suggested by Nicolas et al. (2017).





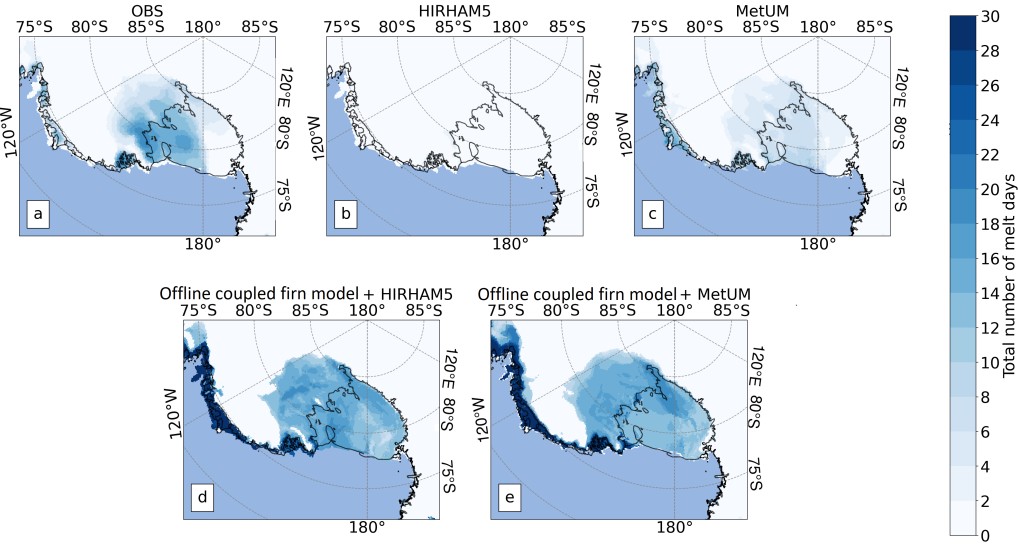

**Figure 2.** Maps of West Antarctica showing the total number of melt days (shading) during January 2016 from (a) satellite passive microwave measurements, (b) native melt output from HIRHAM5, (c) native melt output from MetUM, (d) melt output from the offline coupled firn model forced by HIRHAM5 output, and (e) melt output from the offline coupled firn model forced by MetUM output.

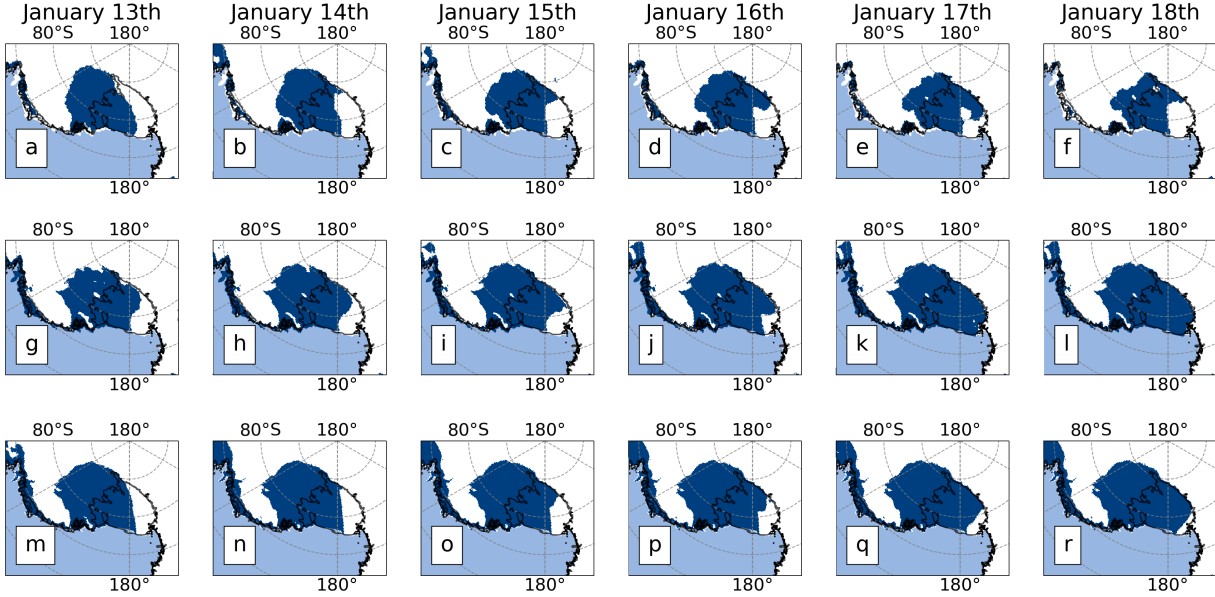

**Figure 3.** Maps of West Antarctica showing the daily melt area from the 13[th] to 18[th] (from left to right) of January 2016 from (top row; a-f) satellite passive microwave measurements, (middle row; g-l) the offline coupled firn model forced by HIRHAM5 output, and (bottom row; m-r) the offline coupled firn model forced by MetUM output. Melt areas are indicated by the dark shading, while melt-free regions are shown as white.

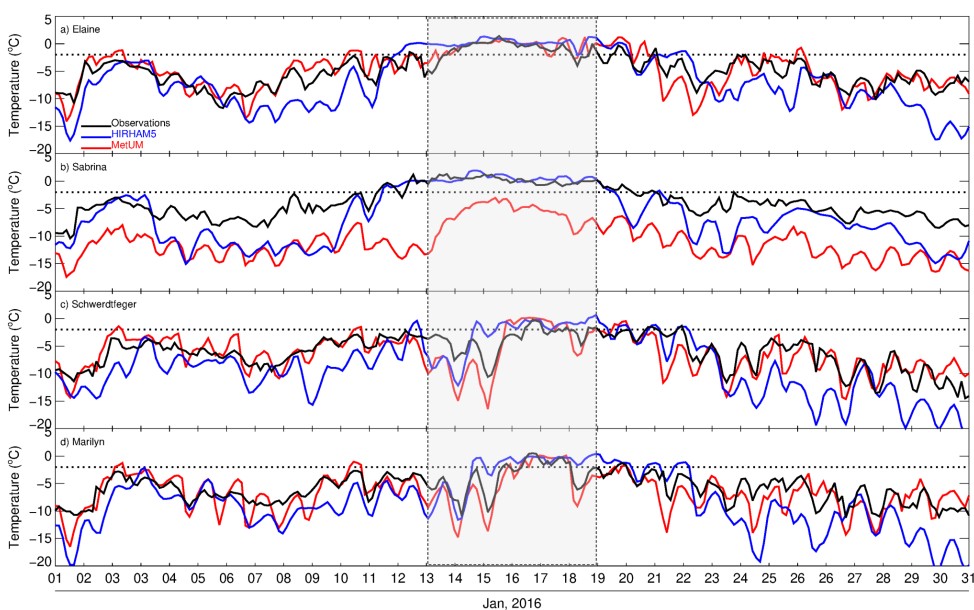

**Figure 4.** Timeseries of near-surface air temperature (°C) from the 1st to 31st of January 2016 from AWS measurements (black line), HIRHAM5 output (blue line), and MetUM output (red line) at (a) Elaine, (b) Sabrina, (c) Schwerdtfeger, and (d) Marilyn. The period of special interest from 13th to 18th of January is highlighted by the semi-transparent shaded region. The date shown is in UTC, with local time for the Ross Ice Shelf 12 hrs ahead of UTC. The horizontal dotted line shows temperatures at -2°C, which Nicolas et al. (2017) suggests is the threshold for melting for this particular event.

## 4 Cloud radiative effects

The firn model simulation of daily melt extent over the western sector of the RIS is broadly in agreement with the satellite-based observations from the 13th to 15th and then in disagreement from the 16th to 18th (Fig. 3). To investigate this, Fig. 5 compares
the timeseries for this period of surface radiative fluxes that are spatially-averaged over the western sector of the RIS (region highlighted in Fig. 1) from HIRHAM5, MetUM, and CERES. Model estimates of SEB are also included, although these are not available for CERES. For the initial part of the timeseries from the 13th to 14th, the diurnal cycle of net surface radiative flux shows negative values during nighttime of around -20 to -40 W m$^{-2}$ for HIRHAM5 and around -20 W m$^{-2}$ for MetUM, which are broadly consistent with freezing and thus with the firn models correctly simulating this region as being melt-free. Note
that the models also show positive net surface radiation values during daytime, suggesting a daily freeze-thaw cycle. However, from the 15th to 18th the models simulate a transition towards values of nighttime/minimum net surface radiation flux of around zero, which is broadly consistent with an absence of freezing and the firn models (erroneously) simulating surface melt. It is





also apparent that although the minimum net surface radiation is broadly similar for both models, the daytime maximum net surface radiation values are larger for HIRHAM5 compared to MetUM, e.g., for the 16[th], 17[th], and 18[th] the HIRHAM5 values are around 50, 10 and 20 W m$^{-2}$ larger than those of MetUM. This is possibly consistent with the HIRHAM5-forced firn model estimate of melt extent being much more (erroneously) extensive over the western RIS compared to the MetUM-forced estimate (Fig. 3), as well as HIRHAM5 simulating warmer near-surface temperatures over this region compared to MetUM (Fig. 4). Note that the diurnal cycle of SEB from the models broadly follows the net surface radiative flux cycle. For example, the nighttime/minimum SEB values from the models shows negative values for the initial part of the timeseries and zero/positive values for the later part of the timeseries. This suggests that the primary energy source responsible for the transition in SEB is from surface radiative flux and not sensible and latent heat fluxes (Nicolas et al., 2017).

Examination of the timeseries of net surface SW and net surface LW fluxes from HIRHAM5 and MetUM in Fig. 5 suggests that the radiative fluxes are finely balanced with respect to surface melt. In particular, the transition from negative values of nighttime/minimum net surface radiation flux to zero/positive values is mainly due to net surface LW values becoming less negative. For example, values of net surface LW flux change from -90 to -100 W m$^{-2}$ on the 14[th] and 15[th] to around -20 W m$^{-2}$ on the 17[th] and 18[th]. By contrast, the net surface SW values show little change in either direction at night (as expected), i.e., they are unable to offset the changes in net surface LW flux. Figure 5 further shows that the changes in net surface LW values are due to a marked increase in surface downwelling LW flux, which increases from around 180 to 200 W m$^{-2}$ on the 14[th] and 15[th] to around 280 W m$^{-2}$ on the 17[th] and 18[th].

Interestingly, Fig. 5 results also show that the CERES estimates of the nighttime/minimum net surface radiation flux are 20-50 W m$^{-2}$ larger than the model estimates. Moreover, CERES values are positive during both night and daytime from the 13[th] to 18[th]. This raises concerns over the reliability of these measurements, as this would also presumably be associated with (erroneous) melt over the western RIS region, i.e., contradicting the satellite passive microwave measurements of daily melt extent (Figs. 2 and 3). The comparison also shows that the CERES positive net surface SW fluxes are considerably greater than the model estimates, i.e., consistent with CERES estimating a higher net radiation flux compared to the models. The CERES estimates of net surface LW fluxes show the same transition to smaller negative values from the 15[th] to the 17[th] that is apparent in the models. However, the CERES estimates of surface downwelling LW flux are around 30 W m$^{-2}$ larger than the model values for the initial part of the timeseries, and then lower than the model values during the later part of the timeseries when a smaller increase from 220 W m$^{-2}$ to around 260 W m$^{-2}$ occurs.

To further understand the discrepancies in melt area over the western RIS region, Figs. 6, 7, and 8 show the spatial distributions of net surface radiative flux, net surface LW flux, and surface downwelling LW flux, respectively, from HIRHAM5, MetUM, and CERES at 12 UTC on the 14[th] of January and 12 UTC on the 17[th] of January, i.e., representative of nighttime conditions as the local time for the Ross Ice Shelf is 12 hrs ahead of UTC. Figure 6 shows that the negative net surface radiative flux values simulated by HIRHAM5 and MetUM over the western RIS during nighttime on the 14[th] are actually largely constrained to this region and do not extend over the rest of the RIS. Over this region the values are around -35 W m$^{-2}$ for HIRHAM5 and slightly smaller for MetUM (c.f., Fig. 5). By contrast, over the central and eastern sectors of the RIS the simulated values of net surface radiative flux are mostly positive, i.e., consistent with the firn model simulating melting here, in





agreement with the observations. Figure 6 also shows that the region of weakly positive net surface radiative values (around zero) simulated by the models during nighttime on the 17[th] (c.f., Fig. 5) actually extends over the entire western sector of the

RIS, i.e., the sector bordering the entire length of the Transantarctic Mountains. By contrast, the simulated net surface radiative values over the eastern and central sectors of the ice shelf are largely negative during nighttime on the 17[th]. As shown in Fig. 5, CERES estimates of net surface radiative flux are mainly positive over most of the RIS during both the 14[th] and 17[th], which is in disagreement with the models.

Figure 7 shows that the large negative net surface LW fluxes of up to -100 W m[-2] simulated by HIRHAM5 and MetUM

over the western sector of the RIS during nighttime on the 14[th] are also largely constrained to this region (c.f. Fig. 5), with the negative values over the central and eastern sectors of the ice shelf considerably smaller compared to the western region. The results also confirm the transition to much smaller negative values of net surface LW fluxes during nighttime from the 14[th] to the 17[th] over this region (c.f. Fig. 5). However, over the central and eastern sectors of the RIS during nighttime on the 17[th] the simulated negative net surface LW fluxes are markedly larger than the values over the western sector. Figure 7 also shows that

CERES semi-captures the transition from large negative net surface LW values over the western RIS during nighttime on the 14[th] to smaller negative values on the 17[th], in agreement with the models.

The marked increase in surface downwelling LW flux simulated by the models over the western sector of the RIS during nighttime is confirmed in Fig. 8, with values around 200 W m[-2] on the 14[th] and around 280 W m[-2] on the 17[th] (c.f. Fig. 5). Also apparent is that the surface downwelling LW flux over the central and eastern sectors of the RIS during nighttime on the 14[th] is

considerably larger compared to the western sector, which is consistent with Fig. 7. Figure 8 confirms that CERES estimates of surface downwelling LW flux are considerably higher than the models estimates on the 14[th]. Note that examination of the net surface SW flux simulated by the models showed broadly similar values on the 14[th] and the 17[th] over the western RIS during nighttime (not shown).



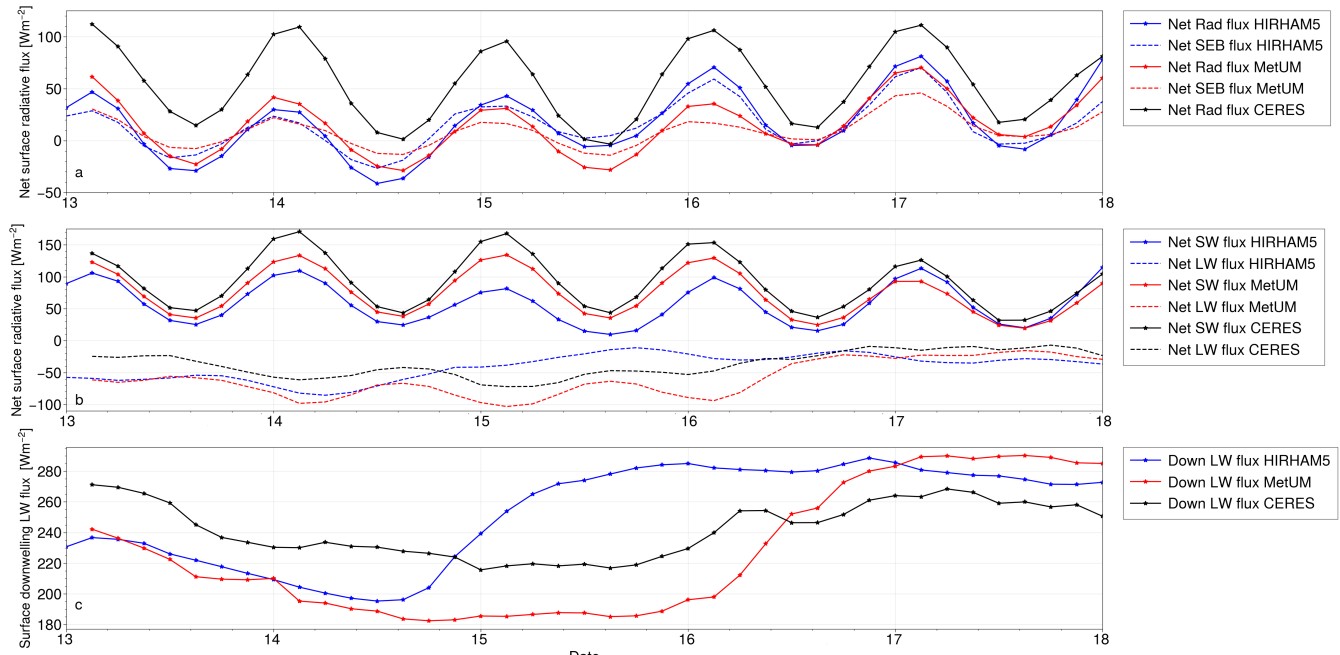

**Figure 5.** Timeseries of surface radiative fluxes from the 13<sup>th</sup> to 18<sup>th</sup> of January 2016 that are spatially-averaged over the western sector of the RIS (region highlighted in Fig. 1) from HIRHAM5 and MetUM simulations and the CERES observations (W m$^{-2}$). Panel (a) shows the net surface radiative fluxes and SEB. Panel (b) shows the net surface LW and SW fluxes. Panel (c) shows the surface downwelling LW fluxes. The date shown is in UTC, with local time for the Ross Ice Shelf 12 hrs ahead of UTC. Note that SEB measurements are not available from CERES.



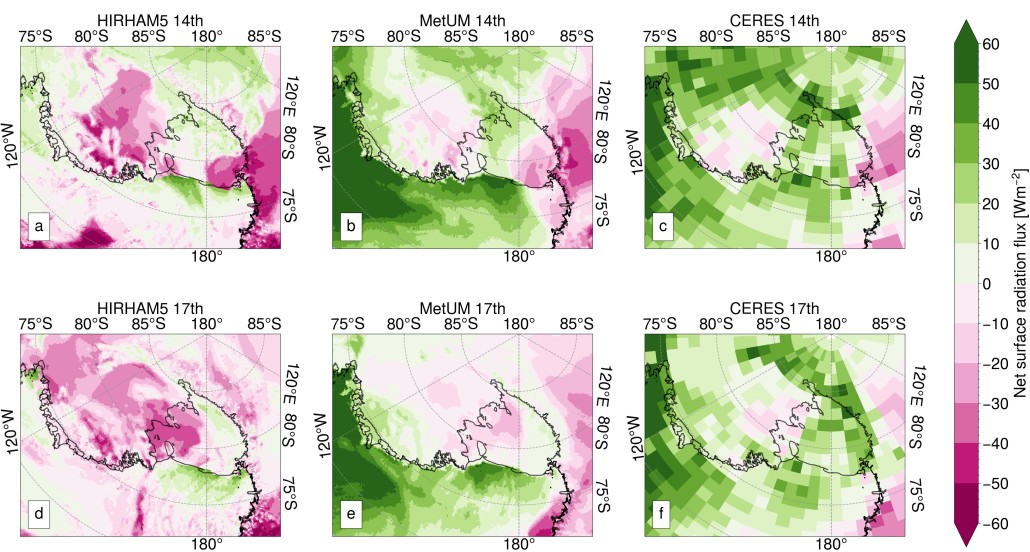

**Figure 6.** Maps of West Antarctica showing 3-hourly averaged net surface radiative fluxes (W m$^{-2}$) at 12 UTC on the 14$^{th}$ of January 2016 (top row; a-c) and 12 UTC on the 17$^{th}$ of January 2016 (bottom row; d-f) from HIRHAM5 (left column; a, d), MetUM (middle column; b, e), and CERES (right column; c, f). Downward fluxes are positive. Note that 12 UTC is equivalent to 00 LT over the Ross Ice Shelf.

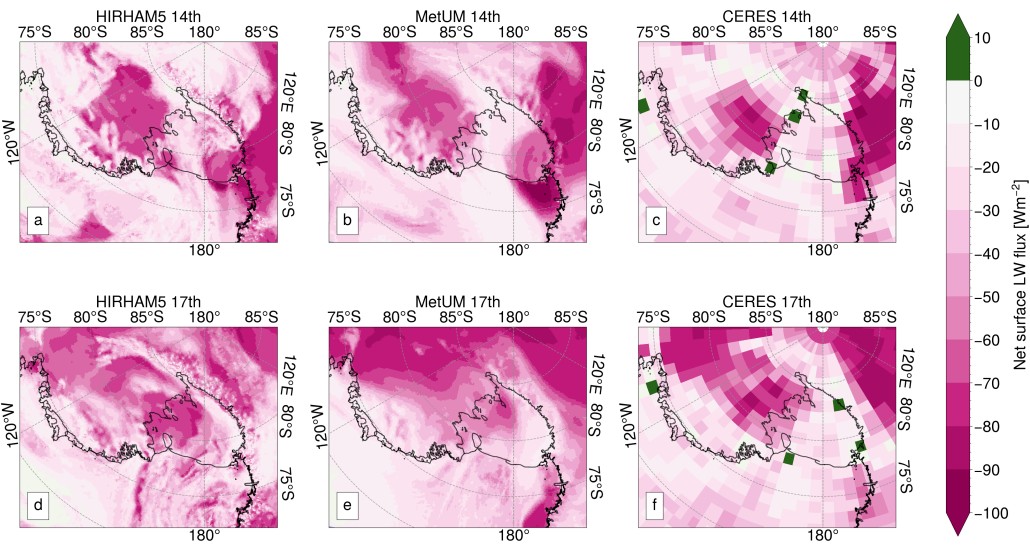

**Figure 7.** Maps of West Antarctica showing 3-hourly averaged net surface LW fluxes (W m$^{-2}$) at 12 UTC on the 14$^{th}$ of January 2016 (top row; a-c) and 12 UTC on the 17$^{th}$ of January 2016 (bottom row; d-f) from HIRHAM5 (left column; a, d), MetUM (middle column; b, e), and CERES (right column; c, f). Downward fluxes are positive. Note that 12 UTC is equivalent to 00 LT over the Ross Ice Shelf.

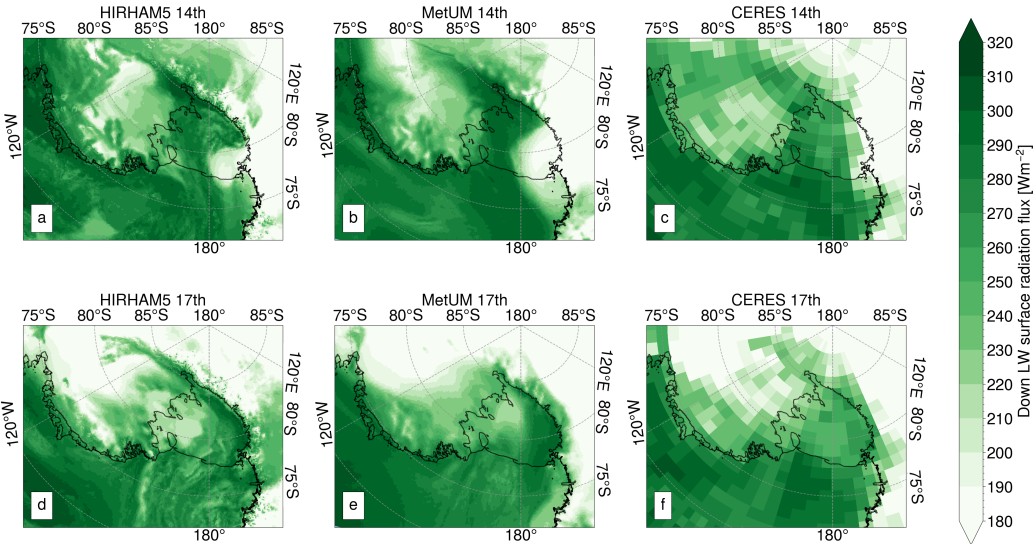

**Figure 8.** Maps of West Antarctica showing 3-hourly averaged surface downwelling LW fluxes (W m$^{-2}$) at 12 UTC on the 14$^{th}$ of January 2016 (top row; a-c) and 12 UTC on the 17$^{th}$ of January 2016 (bottom row; d-f) from HIRHAM5 (left column; a, d), MetUM (middle column; b, e), and CERES (right column; c, f). Downward fluxes are positive. Note that 12 UTC is equivalent to 00 LT over the Ross Ice Shelf.

## 5 Cloud properties

Figures 9 to 12 compare cloud properties between the 14$^{th}$ and the 17$^{th}$ of January to help explain the differences in radiative fluxes (Figs. 5 to 8) over the western sector of the RIS. Figure 9 shows the low-level cloud fraction from HIRHAM5 and MetUM at 12 UTC on the 14$^{th}$ January and 12 UTC on the 17$^{th}$ January, i.e., the same times as examined in Figs. 6 to 8. Also shown are MODIS observations at these times. On the 14$^{th}$ of January, the models show largely cloud-free conditions over the western RIS region (0%), in contrast to extensive cloud over the melting-areas of the eastern and central sectors of the RIS

(>80%). However, on the 17$^{th}$ of January the simulations show extensive cloud cover over the western and central sectors of the RIS and more cloud-free conditions over the eastern sector. Note that the HIRHAM5 and MetUM simulated medium- and upper-level cloud fractions showed cloud-free conditions during these times (not shown). The model estimates of cloud cover on both the 14$^{th}$ and 17$^{th}$ are largely in agreement with the MODIS imagery, especially over the western RIS region - although MODIS shows a slightly smaller cloud-free area over this region on the 14$^{th}$ compared to the models.

The good agreement between simulated and observed cloud cover over the western RIS (Fig. 9) suggests that the potential misrepresentation of surface melting/net surface LW/surface downwelling LW fluxes may stem from factors beyond simple cloud cover, to other processes such as cloud phase. This can be explained by Fig. 10, which shows the vertical profile of cloud phases and their respective heights retrieved by the CALIPSO satellite during its passage over the western RIS at around 06 UTC on 14$^{th}$ and 17$^{th}$ January (see Fig. 1 for the CALIPSO ground tracks) – note that the difference in times compared to Fig.

9 makes a direct comparison difficult. Nevertheless, CALIPSO detected mostly liquid-based clouds between 2-4 km above the



RIS on 14$^{th}$ of January, including this western sector (Fig. 10). Although it is noteworthy that the satellite track over this region was relatively far north, and at 06 UTC the MODIS imagery showed cloudy conditions over the north-western sector of RIS (not shown). However, as mentioned above, MODIS imagery confirms that the western sector of the RIS can be considered as largely cloud-free on the 14$^{th}$ (Fig. 9). More noteworthy is that CALIPSO shows liquid-water and ice-water clouds extending

up to 7 km above the surface in the same region on the 17$^{th}$ of January (Fig. 10) coincident with the (erroneous) spike in modelled melt.

As seen by Fig. 11, the cloud ice water path and liquid water path simulated by the MetUM, shows negligible values of cloud liquid water and ice water content over the western sector of the RIS on the 14$^{th}$ of January, which is consistent with it also simulating cloud-free conditions over this area (Fig. 9). However, on the 17$^{th}$ January the MetUM simulates much higher

amounts of cloud ice-water over the western sector of the RIS compared to cloud liquid-water, with a maximum cloud ice water path of around 0.5 kg m$^{-2}$. Figure 12 shows equivalent results for CERES, which also indicate negligible values of cloud ice/liquid water path at 12 UTC on the 14$^{th}$ of January over the western sector of the RIS, in agreement with the MetUM. However, CERES suggests that clouds with high liquid-water content and ice-water content occur at 12 UTC on the 17$^{th}$ over this region, with values of cloud ice water path up to 0.5 kg m$^{-2}$ (i.e., similar to the MetUM) and cloud liquid water path up

to 1 kg m$^{-2}$ (i.e., two orders of magnitude larger than the MetUM). Moreover, it's noteworthy that CALIPSO also observed liquid-water and ice-water clouds over the western region of the RIS (Fig. 10), which substantiates the CERES results.





**Figure 9.** Maps of West Antarctica showing cloud cover (%) at 12 UTC on the 14[th] of January 2016 (top row; a-c) and 12 UTC on the 17[th] of January 2016 (bottom row; d-f) from HIRHAM5 (left column; a, d), MetUM (middle column; b, e), and MODIS (right column; c, f). The model results are based on 6-hourly averages of low-level cloud fraction. For MODIS, a pseudo-image at 12 UTC was calculated by combining MODIS imagery that corresponded to satellite ground tracks over the western sector of the RIS at around 06 UTC and 18 UTC and then averaging. Note that 12 UTC is equivalent to 00 LT over the Ross Ice Shelf.







**Figure 10.** Observed vertical profile of cloud phase from the CALIPSO satellite over the RIS at around 06 UTC on a) the 14<sup>th</sup> of January and b) the 17<sup>th</sup> of January. The blue bars at the bottom of each panel highlight the part of each satellite ground track that is over the western region of the RIS highlighted in Fig. 1. Note that 06 UTC is equivalent to 18 LT over the Ross Ice Shelf.



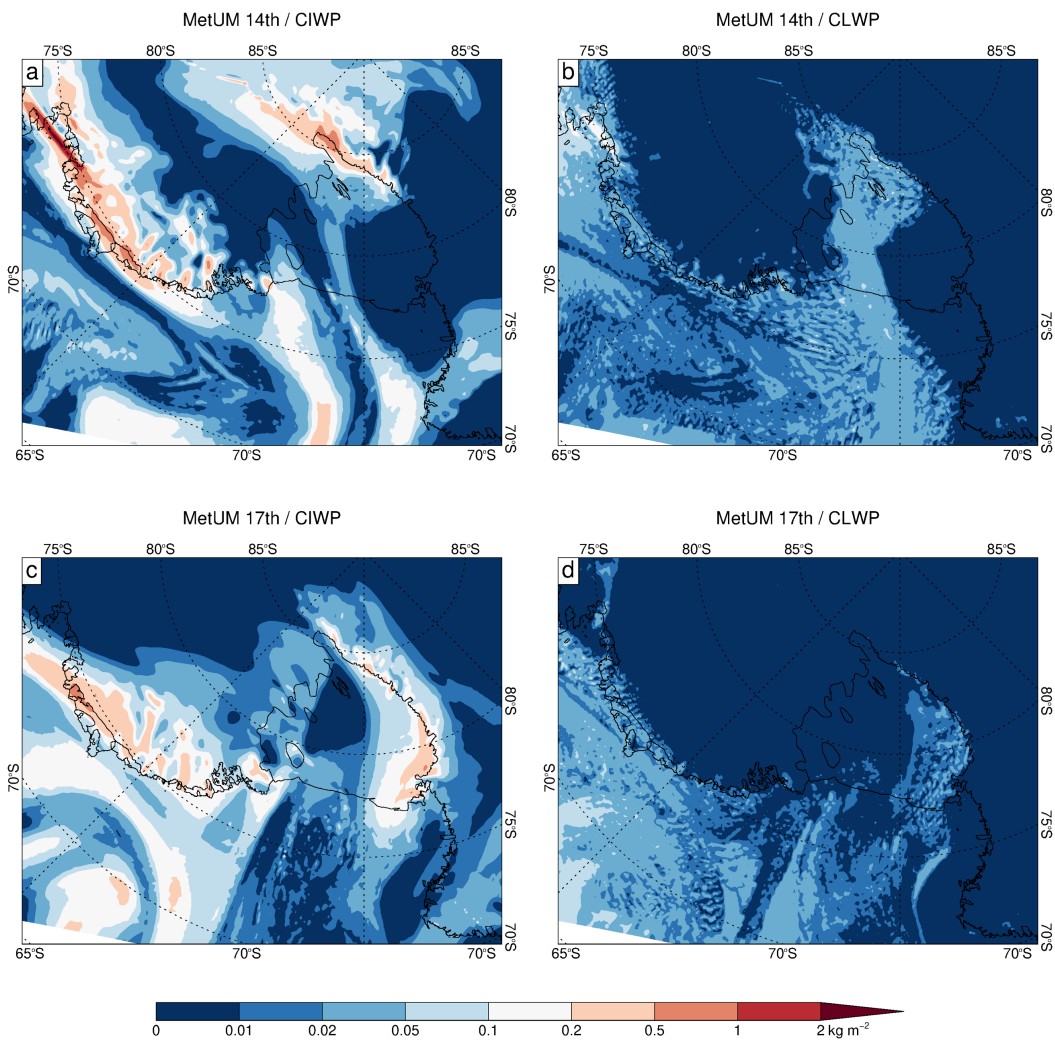

**Figure 11.** Maps of West Antarctica showing the cloud ice water path (CIWP; left column; a, c; kg m$^{-2}$) and cloud liquid water path (CLWP; right column; b, d; kg m$^{-2}$) at 12 UTC on the 14$^{th}$ of January 2016 (top row; a, b) and 12 UTC on the 17$^{th}$ of January 2016 (bottom row; c, d) from the MetUM, based on instantaneous values. Note that equivalent cloud ice/liquid water path information from the HIRHAM5 simulation was not available. Note also that 12 UTC is equivalent to 00 LT over the Ross Ice Shelf.




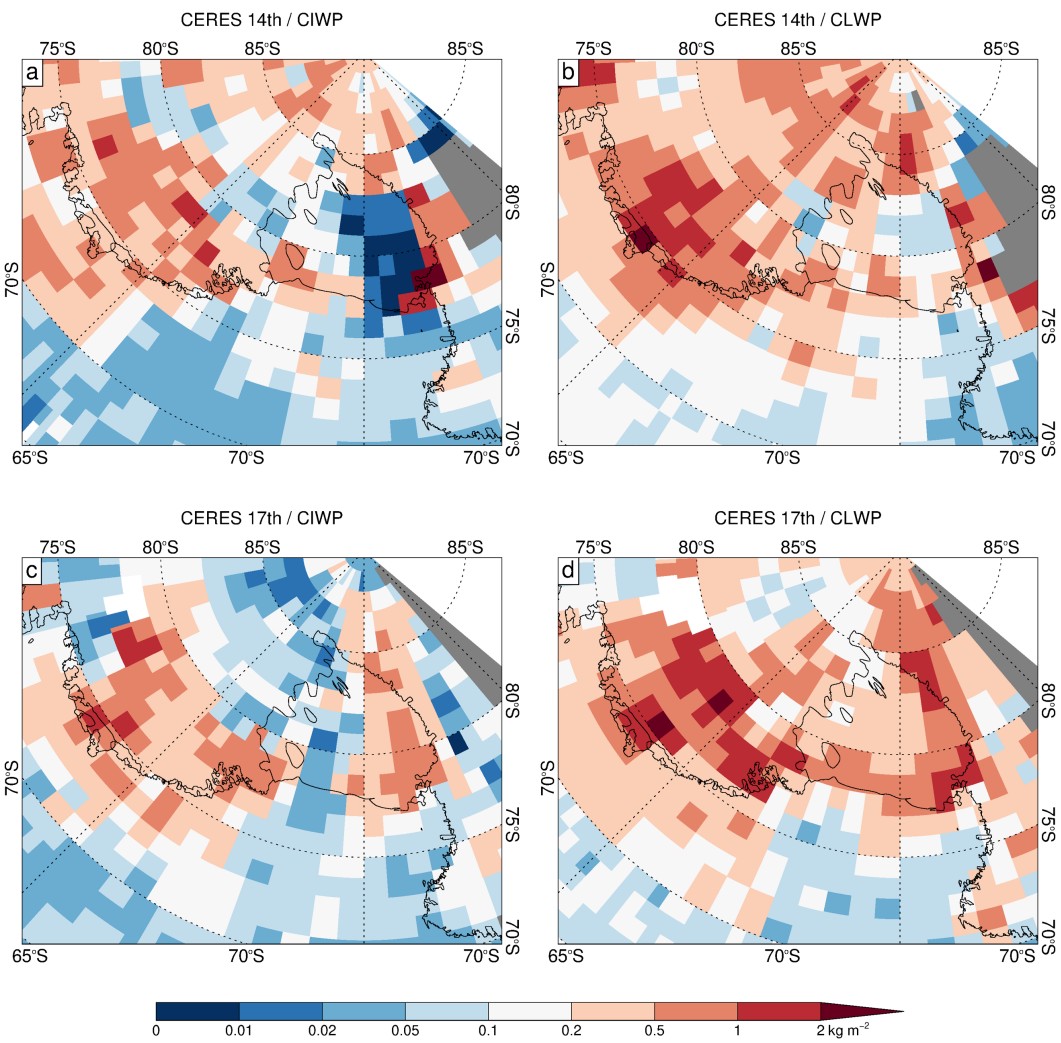

**Figure 12.** Maps of West Antarctica showing the cloud ice water path (CIWP; left column; a, c; kg m$^{-2}$) and cloud liquid water path (CLWP; right column; b, d; kg m$^{-2}$) at 12 UTC on the 14$^{th}$ of January 2016 (top row; a, b) and 12 UTC on the 17$^{th}$ of January 2016 (bottom row; c, d) from CERES observations, based on 3-hourly averages. Note that 12 UTC is equivalent to 00 LT over the Ross Ice Shelf.

# 6 Discussion

The distinct melt-free region that is observed over the western sector of the RIS from the 16$^{th}$ to the 18$^{th}$ of January (Fig. 3) coincides with observations from MODIS showing cloudy conditions (Fig. 9). Observations from CALIPSO and CERES show that these cloud conditions are characterised by both liquid-water and ice-water clouds (Figs. 10 and 12). In addition, the "unknown" classification used by CALIPSO likely indicates mixed-phase clouds, which are therefore also apparent (Fig. 10).



However, although both HIRHAM5 and MetUM capture the cloudy conditions over the western sector of the RIS on the 17[th], it is highly likely that they have deficiencies in the representation of the ice-to-liquid partitioning of cloud water (Fig. 11). For this event, the MetUM simulates much higher amounts of cloud ice-water over the western sector of the RIS compared to cloud
liquid-water (Fig. 11), which is in disagreement with both CALIPSO and CERES observations that suggest that both phases are important (Figs. 10 and 12). We therefore suggest that model deficiencies in both the MetUM and HIRHAM5 cloud phase partitioning are likely to be at least partly responsible for the errors in simulated LW radiative fluxes, as these are known to be highly sensitive to this (Zhang et al., 1996; Gilbert et al., 2020; Ghiz et al., 2021). Unfortunately, there are no AWS radiation measurements available to explicitly assess these errors.

The enhancement of surface downwelling LW radiation simulated by HIRHAM5 and MetUM is consistent with the models simulating much higher amounts of optically thick ice-water cloud, as this would contribute a LW warming effect (Zhang et al., 1996). However, the larger amounts of liquid-water clouds observed by CALIPSO and CERES would be expected to produce even larger downwelling surface LW fluxes (Zhang et al., 1996). This is not the case, suggesting that other factors influencing the LW radiative effect of the clouds, such as cloud temperature, altitude, and cloud microphysical properties like
the size of water droplets or ice crystals, may be impacting surface LW fluxes. In reality, multiple possible cloud properties (in addition to ice-to-liquid partitioning of cloud water) could be influencing the radiative effects of the clouds to produce smaller downwelling LW fluxes than are being simulated. This would likely result in larger negative values of nighttime/minimum net surface radiation flux (and therefore larger negative SEB values) over the western sector of the RIS during this period and explain why this region was observed to be melt-free.

Addressing such deficiencies in cloud schemes will require increasing the number of well-instrumented Antarctic stations that are able to make comprehensive measurements of radiation and clouds (Lubin et al., 2020; Zhang et al., 2023). Efforts are underway to improve the surface observing network to have full four-component radiation measurements, which will also require additional care to ensure the measurements are of sufficient quality to be used in future studies. Repeating a study such as this for a melt event that includes such measurements would be worthwhile, as would repeating the MetUM simulations
using its recently developed double-moment microphysics scheme to examine whether this increased the amount of liquid-water cloud and limited its conversion to ice (Field et al., 2023). Additionally, more information on the phase of cloud in Antarctica and their vertical structure using flight campaigns are also required. Novel attempts to measure the amount of cloud liquid-water and ice-water using radiosondes have recently been developed and are suitable for use in Antarctica (Smith et al., 2019).

Previous studies have already shown that the MetUM has deficiencies in its representation of cloud phase, particularly related to it simulating Antarctic clouds that contain too much ice-water content and not enough liquid-water content (Abel et al., 2017). Other regional atmospheric models also struggle with correctly partitioning cloud phase in such complex regimes (Bodas-Salcedo et al., 2012; Abel et al., 2017; Hyder et al., 2018; Gilbert et al., 2020). Moreover, the representation of cloud properties in general is a long-standing problem in both regional and global atmospheric models (Van Wessem et al.,
2014, 2018; Lenaerts et al., 2017b; Hines et al., 2019). For example, the parameterisation of sub-grid scale cloud processes and cloud phase partitioning is improved by using higher resolution, but significant biases remain in global and regional models



that contribute to surface energy balance biases, particularly over the Southern Ocean (Bodas-Salcedo et al., 2014; Schudde-boom and McDonald, 2021) but also over ice shelves (King et al., 2015; Gilbert et al., 2020). Additionally, Gilbert et al. (2020) highlighted the need to improve phase partitioning throughout the vertical profile, as poor representation of cloud layers can

also create biases considerable enough to affect surface melting.

The impact of cloud properties on melt has also been studied in Greenland and over Arctic sea ice, where similar biases in atmospheric models have been identified (Van Tricht et al., 2016; Lenaerts et al., 2017a, 2020; Huang et al., 2019). Improving cloud parameterisation schemes will therefore likely improve the representation of Arctic as well as Antarctic melt.

## 7   Conclusions

This study examines the representation of an extensive melt event that occurred over the RIS during January 2016 by the HIRHAM5 and MetUM high-resolution regional atmospheric models, as well as a physically-based, multi-layer, offline coupled firn model forced by both HIRHAM5 and MetUM output. The results show that both the HIRHAM5 and MetUM simulations considerably underestimated the number of melt days that occurred during the event, which is likely due to both limitations in their own ice/snow surface schemes and an absence of spin-up. However, using HIRHAM5 and MetUM output

to force the offline coupled firn model resulted in a considerable improvement in modelled melt. Although the firn model represents the firn layer in a sophisticated manner, including processes such as meltwater percolation, retention, and refreezing, the considerable improvement in the simulation of the melt event by this model is also likely due to it being adequately spun up to ensure a realistic representation of snow and firn properties. However, despite its sophistication, the offline coupled firn model was unable to realistically represent the complete melt pattern over the RIS, and in particular the distinct melt-free region that

occurs over the western sector of the RIS from the 16[th] to the 18[th] of January.

We speculate that the sustained melting over the western sector of the RIS that is wrongly simulated by the firn model originates from the HIRHAM5 and MetUM output used for forcing. In particular, both models erroneously simulate zero/positive values of nighttime/minimum net surface radiation flux (and associated SEB) over the western sector of the RIS during this period, which is broadly consistent with an absence of freezing. This occurs in response to the models simulating a considerable

increase in surface downwelling LW flux from around 180 to 200 W m$^{-2}$ to around 280 W m$^{-2}$ over the course of a few days, leading to an excessive amount of energy at the surface available for melt. Our results suggest that deficiencies in cloud phase partitioning by HIRHAM5 and MetUM are likely to be partly responsible for the misrepresentation of surface downwelling LW flux/surface melting over the western RIS, and not deficiencies in their representation of cloud cover.

This study emphasises the complexity of the processes governing ice shelf melt, and the need for further detailed in-situ

measurements of radiative flux and cloud properties over Antarctic ice shelves to better understand these processes and improve their representation in models. It particularly highlights the urgent need for improvement in the representation of cloud phase partitioning in models.



*Data availability.* The MODIS, CERES, and CALIPSO data used in this study are available from NASA (National Aeronautics and Space Administration). The MODIS data are available here: https://modis.gsfc.nasa.gov/data/dataprod/mod06.php. The CERES data are available here: https://ceres-tool.larc.nasa.gov/ord-tool/jsp/SYN1degEd41Selection.jsp. The CALIPSO data are available here: https://eosweb.larc.nasa.gov/clouds). The AWS data are available here for: a) Sabrina https://doi.org/10.48567/y3s5-3864, b) Elaine https://doi.org/10.48567/tytb-dk68, c) Marilyn https://doi.org/10.48567/kxn6-6246, and d) Schwerdtfeger https://doi.org/10.48567/96v9-mz68. The model output from the HIRHAM5, MetUM, and offline coupled firn model simulations are available here: https:\doi.org/10.5281/zenodo.8355571

*Author contributions.* NH and AO conceived the study. NH, AO, and FB ran the model simulations. NH, AO, XZ, and TP analysed the data. NH, AO, and XZ wrote the initial draft. TB, EG, PL, RM, RP, SW, ML, and SS contributed to analysis of the results and reviewing the writing.

*Competing interests.* At least one of the (co-)authors is a member of the editorial board of The Cryosphere.

*Acknowledgements.* AO, RM, EG, RP, and PM received support from the European Union's Horizon 2020 research and innovation framework programme under Grant Agreement 101003590 (PolarRES). AO and TB received support from the Natural Environment Research Council (NERC) National Capability International grant SURface FluxEs In AnTarctica (NE/X009319/1). XZ received support from National Science Foundation (NSF) Grant 2229392. PL gratefully acknowledges the financial contributions of Aarhus University Interdisciplinary Centre for Climate Change (iClimate, Aarhus University). ML received support from the US National Science Foundation Grants 1924730 and 1951603. Additional work by NH and RM is supported by the Danish State through the National Centre for Climate Research (NCKF), furthermore, NH and RM are supported by the Novo Nordisk Foundation project PRECISE (NNF23OC0081251).





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
