# Peer review of "The importance of cloud properties when assessing surface melting in an offline coupled firn model over Ross Ice shelf, West Antarctica"

_The Cryosphere, 2023_

## Author Comment (AC1)

Hansen et al 2023 doi.org/10.5194/tc-2023-145

[Figure]
 Danish Meteorological Institute

Reply to reviewer comments on

**"The importance of cloud phase when assessing surface melting in an offline coupled firn model over Ross Ice shelf, West Antarctica"**

by

Nicolaj Hansen, Andrew Orr, Xun Zou, Fredrik Boberg, Thomas J. Bracegirdle, Ella Gilbert, Peter L. Langen, Matthew A. Lazzara, Ruth Mottram, Tony Phillips, Ruth Price, Sebastian B. Simonsen, and Stuart Webster

Dear reviewers
On behalf of my co-authors and myself, I would like to thank you for your comments on our manuscript. You have made an extensive review of the manuscript, and we have followed your suggestions to our best efforts. We sincerely believe that your reviews have improved the manuscript.
In the following, we provide a point-by-point answer to all the issues raised by you. All issues will be followed by our suggestions for improvements to the manuscript highlighted in red.

Best regards,
Nicolaj Hansen

Hansen et al 2023 doi.org/10.5194/tc-2023-145

[Figure]

Danish Meteorological Institute

**Reviewer 1:**

My concerns are the following:

- **The two major observational datasets used in the article are contradictory with respect to melt in western RIS** : passive microwave melt extent shows no melt in western RIS during the event, whereas CERES radiative fluxes would lead to more melt than in models, as indicated by the authors in Section 4. Indeed, CERES shows larger net surface radiative fluxes than in models (Fig5a), consistent with larger liquid water path in CERES than in MetUM in western RIS during the event (Fig 11 and 12).

  Thanks for the comment. We have removed CERES data from figures 5-8. This has been done because CERES measures fluxes and cloud properties at the top of the atmosphere. So the surface fluxes are a derivative of the top of the atmosphere observations, Futhermore, Hinkelman and Marchand (2020) suggested a potential positive bias in SW radiation and a negative bias in LW radiation over the Southern Ocean. However, we keep figure 11 and 12 as it is a more direct observation.

[Figure]

Hansen et al 2023 doi.org/10.5194/tc-2023-145

Danish Meteorological Institute

[Figure]

Hansen et al 2023 doi.org/10.5194/tc-2023-145

[Figure]

- The manuscript shows that models simulate too few liquid clouds with respect to ice clouds, using both CERES and CALIPSO satellite products. This means that **correcting the ice-to-liquid mass partitioning would induce more downward longwave radiation toward the surface, hence more melt**. As the authors rely on the passive microwave melt extent rather than on CERES (point above), **this result contradicts the statement in the introduction that** *"The models (…) seemingly struggle to correctly represent the ice-to-liquid mass partitioning associated with the cloudy conditions, which we suggest is responsible for the radiative flux errors"*. This statement is repeated in the discussion, where a large emphasis is given on liquid-ice cloud partitioning. But in the discussion, the authors also state that fixing this partitioning would actually increase melting, in contradiction with passive microwave observations: *"However, the larger amounts of liquid-water clouds observed by CALIPSO and CERES would be expected to produce even larger downwelling surface LW fluxes (Zhang et al., 1996). This is not the case, suggesting that other factors influencing the LW radiative effect of the clouds, such as cloud temperature, altitude, and cloud microphysical properties like the size of water droplets or ice crystals, may be impacting surface LW fluxes."*. **So why this emphasis on ice-liquid partitioning, which contradicts the results of the study?**
- From the two points above, I think that a conclusion of this study is that either **(1) properties other than ice-to-liquid partitioning of cloud water influence the radiative effects of the clouds**, or that **(2) melt extent from passive microwave might be wrong.** Can the authors clarify these points?

Thank you for these comments. We do trust the passive microwave data, as it has been validated multiple times. We will make sure that this is clear in the conclusion. We agree with your conclusion that the ice-liquid partitioning cannot be solely responsible for the radiative flux biases, since correcting it in line with satellite cloud observations would adjust the downwelling LW fluxes in the wrong direction relative to what is required for agreement with passive microwave melt observations. Rather, we intended to emphasise that the incorrect ice-liquid phase partitioning in the simulated clouds is evidence of broader problems with the cloud microphysics schemes. Model deficiencies affecting the ice-liquid partitioning can be expected to impact other cloud properties as well, such as cloud height and temperature, or the vertical distribution of ice relative to liquid within the cloud. These other variables will have an impact on the surface radiative fluxes as well as the simple phase partitioning. So we hypothesise that cloud microphysics, including but not limited to phase, is biased in the model and that this is responsible for the radiative fluxes. We have made several changes to the text, especially in the discussion section, to make our meaning more clear on this subject.

More details are given bellow.

**Specific comments**

Hansen et al 2023 doi.org/10.5194/tc-2023-145

[Figure]

**1 Introduction**

*"Therefore, to realistically capture local climate variability and simulate ice shelf melt patterns, it is essential to utilize regional atmospheric models at high spatial resolution, i.e., grid box sizes of the order 10 km or less."*

- In case of synoptic-scale events, 10 km resolution might not be needed over large ice shelves. It depend on the ice shelves ?
  Thank you for this comment. However, we believe that a 10km resolution is needed as the margin of the ice shelves can show complex wind and temperature pattens.
- Which of the numerous papers cited L25-35 does use "10km or less" resolution ?

  Thank you for this comment. Lenaerts et al., 2017a, Heinemann et al., 2019, Zou et al., 2021, 2023, Gilbert et al., 2022, Bozkurt et al., 2018, and Wille et al., 2022 all have a horinzotal resolution of less than 10 km.

*"Here we investigate the benefits of applying the sophisticated offline coupled firn model described by Langen et al. (2017) that represents key aspects such as the melt-albedo feedback to improve regional atmospheric model simulations of a prolonged and extensive episode of surface melt that occurred during January 2016 over the Ross Ice Shelf (RIS), West Antarctica. The RIS frequently experiences major surface melt events due to both synoptic- and local-scale processes (Nicolas et al., 2017; Zou et al., 2021; Li et al., 2023; Orr et al., 2023), with this particular event attributed to an influx of warm and moist marine air, likely linked to a concurrent strong El Niño episode (Nicolas et al., 2017). The regional atmospheric model simulations examined were initially produced for Antarctic CORDEX (Antarctic COordinated Regional Downscaling EXperiment), and are based on HIRHAM version 5 (HIRHAM5) and MetUM version 11.1 (Orr et al., 2023). In these simulations, HIRHAM5 employed a relatively sophisticated multi-layer snow scheme (Langen et al., 2015), while the MetUM utilized a simple composite snow/soil layer (Best et al., 2011)."*

- This paragraph should be moved to the method section. It could be replaced by a final paragraph in the introduction presenting the outline of the article, with much less detail on the models as they will be presented in the Method section.

  Thank you for this comment. We have removed the model description of the paragraph and put it into the Methods section, this is still in the intro (as we think this outlines the article) :
  *"Here we investigate the benefits of applying the sophisticated offline coupled firn model described by Langen et al. (2017) that represents key aspects such as the melt-albedo feedback to improve regional atmospheric model simulations of a prolonged and extensive episode of surface melt that occurred during January 2016 over the Ross Ice Shelf (RIS), West Antarctica. The RIS frequently experiences major surface melt events due to both synoptic- and local-scale processes (Nicolas et al., 2017; Zouet al., 2021; Li et al., 2023; Orr et al., 2023), with this particular event attributed to an influx of warm and moist marine air, likely linked to a concurrent strong El Niño episode (Nicolas et al., 2017)."*

Hansen et al 2023 doi.org/10.5194/tc-2023-145

[Figure]

**2 Methods and materials**

This section should be divided in (at least) 2 subsections : Observations and Models.

Good idea, we now have a subsection called *Models* and one called *Observations.* On that occasion we have also added some more information on the CALIPSO and CERES data products (suggested by the other reviewer).

*"This consists of 6-hourly averaged values of solid precipitation, liquid precipitation, surface evaporation, surface sublimation, surface downwelling SW radiative flux, surface downwelling LW radiative flux, sensible heat flux, and latent heat flux"*

- Why this models need surface evaporation, surface sublimation **and** latent heat flux as input?
  Thank you for this comment. The model uses evaporation and sublimation to calculate the surface mass balance, the model does not calculate the evaporation and sublimation. The model uses the latent heat flux to calculate the surface energy balance.

*"These are compared with daily melt extent estimates from satellite passive microwave measurements at a grid spacing of 25 km (Picard et al., 2007; Nicolas et al., 2017), using the same melt threshold of 3 mm."*

- I don't find a 3 mm threshold in Picard et al., 2007 nor in Nicolas et al., 2017. Can you justify the choice of this threshold?

  Thank you for the comment, the 3 mm threshold is the Donat-Magnin et al 2020, we will insert the reference.

**4 Cloud radiative effects**

**Comparison with CERES**

*"This raises concerns over the reliability of these measurements, as this would also presumably be associated with (erroneous) melt over the western RIS region, i.e., contradicting the satellite passive microwave measurements of daily melt extent (Figs. 2 and 3)."* And to the end of the section, including Fig 5, 6, 7 and 8

- You state that CERES might give erroneous radiative budget at the surface (Fig 5 + sentence above), so we are not sure if we can trust maps from CERES or not (Fig 6, 7, 8). Consequently, what is the objective of this full section?

  Thank you for this comment. As written above, we have removed CERES from figures 5 to 8. Although we have removed CERES from this section, we still think that this section is nice to keep in the paper, as there are some interesting things happening with the energy fluxes. To accommodate for the changes, we have changed the section title to "Surface radiative fluxes"

Hansen et al 2023 doi.org/10.5194/tc-2023-145

[Figure]

*"Figure 7 also shows that CERES semi-captures the transition from large negative net surface LW values over the western RIS during nighttime on the 14th to smaller negative values on the 17th, in agreement with the models."*

- Do you use CERES to evaluate the models, or do you use the models to validate CERES? With this formulation, it seems that you use the models to validate CERES, which is confusing as the initial objective was to evaluate the models. Can you clarify?

  Thank you for this comment. As written above, we have removed CERES from figures 5 to 8. We will make sure to update the text accordingly.

**5 Cloud properties and 6 Discussion**

From Section 5 and 6, I conclude that **partitioning between liquid and ice cannot be the reason for the supposed too high melt in models versus passive microwave melt extent:**

Thanks for this comment. We agree it is more complex than just the cloud phase as measured earlier.

- **Section 5 : Liquid cloud are observed on western RIS during the event** *"More noteworthy is that CALIPSO shows liquid-water and ice-water clouds extending up to 7 km above the surface in the same region on the 17th of January (Fig. 10) coincident with the (erroneous) spike in modelled melt."* "the same region" being western RIS according to Fig. 10 legend.

  Thank you for this comment. We have changed "liquid-water and ice-water clouds" to "mixed-phase clouds" – the presence of mixed-phase clouds is a more complex regime for the models to simulate, which is why we highlight it.

- **Section 5 : MetUM models much less LWP that CERES and CALIPSO in western RIS during the event** *"However, CERES suggests that clouds with high liquid-water content and ice-water content occur at 12 UTC on the 17th over this region, with values of cloud ice water path up to 0.5 kg m-2 (i.e., similar to the MetUM) and cloud liquid water path up to 1 kg m-2 (i.e., two orders of magnitude larger than the MetUM). Moreover, it's noteworthy that CALIPSO also observed liquid-water and ice-water clouds over the western region of the RIS (Fig. 10), which substantiates the CERES results."*

  Thank you for this comment. We have left this text as it is to highlight the model cloud biases that can be evaluated against the observations.

- **Section 6 : Partitioning between liquid and ice cannot be the reason for discrepancies in LWD.** *"However, the larger amounts of liquid-water clouds observed by CALIPSO and CERES would be expected to produce even larger downwelling surface LW fluxes (Zhang et al., 1996). This is not the case, suggesting that other factors influencing the LW radiative effect of the clouds, such as cloud temperature, altitude, and cloud microphysical properties like the size of water*

Hansen et al 2023 doi.org/10.5194/tc-2023-145

[Figure]

*droplets or ice crystals, may be impacting surface LW fluxes." "In reality, multiple possible cloud properties (in addition to ice-to-liquid partitioning of cloud water) could be influencing the radiative effects of the clouds to produce smaller downwelling LW fluxes than are being simulated."*

Thanks for this comment. We have edited the text related to cloud microphysics in this section. We hope that the new text makes our conclusions clearer.

- From this, I think that a conclusion of this study is that either **(1) properties other than ice-to-liquid partitioning of cloud water influence the radiative effects of the clouds**, or that **(2) melt extent from passive microwave might be wrong.**

Thank you for this comment. We agree that ice-liquid partitioning cannot be solely responsible and have edited the text to emphasise this point.

*"as would repeating the MetUM simulations using its recently developed double-moment microphysics scheme to examine whether this increased the amount of liquid- water cloud and limited its conversion to ice (Field et al., 2023)."*

*"Previous studies have already shown that the MetUM has deficiencies in its representation of cloud phase, particularly re- lated to it simulating Antarctic clouds that contain too much ice-water content and not enough liquid-water content (Abel et al., 2017)."*

- Here, more liquid cloud would induce more melt, so more model bias compared to passive microwave melt extent. Can you clarify what you expect to improve by increasing the liquid water content?

Thank you for this comment. We believe that improving model cloud microphysics schemes would improve cloud properties generally, including phase partitioning but also affect other variables that could be causing biases in the radiative fluxes in this case study.

**Technical corrections**

The number of references L25-35 is too large (25 references)

Thank you for the comment, we have removed some of the references

**Figure 1 :** Orgraphy

Thank you for your comment, we have corrected the typo

Danish Meteorological Institute

[Figure]

**Figure 11 and Figure 12 :** Use a continuous colormap instead of the divergent Blue/Red colormap curenlty used.

Thank you for this comment. We have updated the colormap.

[Figure]

[Figure]

**Reviewer 2:**

Although the fundamental atmospheric result is not entirely new, it is useful to see that applying the most sophisticated firn model does not compensate for the radiative errors related to cloud phase. The manuscript needs some additional detail and clarification regarding some of the satellite remote sensing products.

1.  It is not stated what CERES product is used. There are several available from NASA Langley Research Center (LaRC) and other NASA facilities. CERES does not measure surface radiative fluxes. It only measures top-of-atmosphere radiances over broad spectral intervals, and these are then combined with angular dependence

Hansen et al 2023 doi.org/10.5194/tc-2023-145

models to get top-of-atmosphere fluxes, and then these fluxes are combined with various other satellite data sets and/or radiative transfer models to get estimates of the surface radiation components. Given that CERES is showing such great discrepancy here, it is important to identify which CERES product has been used and discuss potential sources of error with reference to the underlying algorithms (most of which are published by NASA LaRC in the open literature).

Thank you for pointing this out. We have removed the CERES surface fluxes from figures 5-9, and focus on ice- and liquid-water phases from CERES. We have also added more information about the data product in section 2.2 "Observations" and included a discussion about the uncertainties.

2.  Similarly, CALIPSO provides excellent active-sensor detection of cloud vertical extent, but the phase partitioning algorithms have some built-in assumptions and temperature thresholds. The manuscript should give a brief discussion of how the CALIPSO algorithm might lead to uncertainties in what is presented in Figure 10, with the specific vertical temperature profiles over the study domain.

    Thank you for this comment. The CALIPSO dataset distinguishes between ice (depolarizing) and water clouds (spherical) based on backscattered light. However, uncertainties in cloud phase identification can arise from multiple scattering by water clouds, which exhibit significant depolarization, and horizontally oriented ice particles that are nearly nondepolarizing (Hu et al. 2009). Therefore, the vertical profile of cloud phases in Fig. 10 may be influenced by the presence of water clouds, resulting in ice/unknown phases above the surface melting area over the RIS (Fig. 10b). Unfortunately, temperature profile observations over the RIS during the 2016 melt event are unavailable, precluding the provision of additional information on cloud phases.
    We have added some of the above-written text in section 2.2, and in the discussion

3.  The vertical temperature profiles and the vertical profiles of the simulated cloud properties should also be presented and discussed for the two days (14 and 17 January) and the relevant locations. This would make the discussion section (around lines 285-299) less qualitative and speculative. For example, if the temperatures in the lower temperature are only slightly below freezing over several km, then extensive ice phase cloud is obviously ridiculous as we expect supercooled liquid water in these pristine conditions.

    Thank you for this comment. Unfortunately, we have no temperature profiles from CALIPSO or the models.

4.  Regarding the deficiencies of single-moment cloud microphysics, RACMO simulations of West Antarctic surface melt (e.g., see papers by Jan Lenaerts) have "tuned" the microphysical scheme to give high enough cloud liquid water, yielding good geographic representations of surface melt. This should be mentioned somewhere in the later sections of this paper.

[Figure]
 Danish Meteorological Institute

Thanks for pointing this out, we actually already cite papers that deal with tuning/updating cloud schemes, to make that clear we have added this line just before the references
*"and global atmospheric models despite work to improve parameterisations"*

5. Why is ERA-Interim reanalysis used to initialize the regional models and not the more current ERA5?

Thank you for this comment. Neither MetUM nor HIRHAM5 has been run using ERA5 yet.

References:

CERES, N.: CERES and GEO-Enhanced TOA, Within-Atmosphere and Surface Fluxes, Clouds and Aerosols 3-Hourly Terra-Aqua Edition4A, https://doi.org/10.5067/TERRA+AQUA/CERES/SYN1DEG-3HOUR_L3.004A, 2017.

Donat-Magnin, M., Jourdain, N. C., Gallée, H., Amory, C., Kittel, C., Fettweis, X., Wille, J. D., Favier, V., Drira, A., and Agosta, C.: Interannual variability of summer surface mass balance and surface melting in the Amundsen sector, West Antarctica, The Cryosphere, 14, 229–249, https://doi.org/10.5194/tc-14-229-2020, 2020

Hinkelman, L. M. and Marchand, R.: Evaluation of CERES and CloudSat Surface Radiative Fluxes Over Macquarie Island, the Southern Ocean, Earth Space and Science, 7, e2020EA001 224, https://doi.org/10.1029/2020EA001224, 2020

Hu, Y., Winker, D., Vaughan, M., Lin, B., Omar, A., Trepte, C., Flittner, D., Yang, P., Nasiri, S. L., Baum, B., et al.: CALIPSO/CALIOP cloud phase discrimination algorithm, Journal of Atmospheric and Oceanic Technology, 26, 2293–2309, https://doi.org/10.1175/2009JTECHA1280.1, 2009.

---

## Author Response (AR2)

Hansen et al 2023 doi.org/10.5194/tc-2023-145

[Figure]
 Danish Meteorological Institute

Reply to reviewer comments on

**"The importance of cloud properties when assessing surface melting in an offline coupled firn model over Ross Ice shelf, West Antarctica"**

by

Nicolaj Hansen, Andrew Orr, Xun Zou, Fredrik Boberg, Thomas J. Bracegirdle, Ella Gilbert, Peter L. Langen, Matthew A. Lazzara, Ruth Mottram, Tony Phillips, Ruth Price, Sebastian B. Simonsen, and Stuart Webster

Dear Referee #1 and Editor,

Thanks for these follow-up comments. We very much appreciate the expertise and time that has been spent on this task, and sincerely believe that your efforts have resulted in a considerably improved manuscript. We are also very glad that you liked our responses to the first round of comments. We have again implemented your suggested changes / comments as best we can, and very much hope that you now find the manuscript acceptable for publication. In the following, we provide a point-by-point reply to each of these.

Yours sincerely,

Nicolaj Hansen

- I would like to thank the authors for their revision of the article, which addresses many of my comments.
  Thank you for the positive feedback.

- A said in my previous review, this study demonstrates that either (1) properties other than ice-to-liquid partitioning of cloud water influence the radiative effects of the clouds, or that (2) melt extent from passive microwave might be wrong.
  Point (1) was well taken into account in the answers. I find the new wording of the Discussion very clear and convincing.
  Thank you for the positive feedback.

[Figure]

- However, I think it is important for the authors to explore further point (2), namely that the extent of microwave melting could be wrong on 17 (and 18) January 2016:
  We also agree that this should be explored further in the manuscript.  Please see the response to the individual points below for details of how we have gone about this.

- The author should add information on the strength and limitation of microwave-based melt extent in the method section, including under which conditions it is robust or not.
  We very much agree with this comment given the importance of the satellite-based melt results. We have therefore substantially strengthened our description of this in section 2.2, and especially cite a number of publications that confirm that this approach is well suited for detecting surface melting.  The revised text now states:
  *"Microwave remote sensing is particularly suited to detecting surface meltwater over ice shelves because (a) the appearance of liquid water causes an abrupt increase in brightness temperatures and (b) the observations can be acquired during day and nighttime and clear and cloudy conditions (Picard et al., 2007; Nicolas et al., 2017; Johnson et al., 2020, 2022; Mousavi et al., 2022; de Roda Husman et al., 2024). Here, we use a gridded daily surface melting dataset based on Special Sensor Microwave Imager Sounder (SSMIS) satellite-based observations, which uses horizontally polarized brightness temperatures at 19 GHz to identify surface melt. See Nicolas et al. (2017) for further details on the melt detection method. This dataset is available at a spatial resolution of 25 × 25 km, with each grid point classified as either 1(meaning melt was detected during the corresponding day) or 0 (melt not detected)."*

- The authors should discuss how does the choice of the melt threshold (3 mm) affects the comparison with model outputs
  We selected a threshold of 3 mm because this has been used in previous studies to compare satellite-based and model-based melt patterns (e.g., Lenaerts et al. , 2017; Deb et al., 2018; Donat-Magnin et al., 2020).  However, we also conducted an initial sensitivity analysis examining how the simulated melt patterns varied depending on whether a melt threshold of 1, 3 or 5 mm was chosen (see Figures 1-3 below). This analysis actually shows that the patterns of melt extent were largely insensitive to the threshold chosen. To address the comment in the manuscript, the text in section 2.1 was revised to state: *"Note that we found that the modelled patterns of daily melt extent were broadly similar for melt thresholds of 1, 3, and 5 mm per day (not shown), but selected 3 mm per day as this is the same threshold used by Lenaerts et al. (2017b); Deb et al. (2018); Donat-Magnin et al. (2020)."*

- The authors should discuss the robustness of model-data comparison for the 17th and 18th of January, with respect to the 2 points above and the comparison with other datasets. Notably, the 17th of January, all other datasets seem to point to surface melt: high temperature (Figure 4, weather stations, T° > -2) and high liquid water path (Figure 10, CALIPSO track and Figure 12, CERES). The 18th of January, observed surface temperature is bellow -2°C, we don't have a CALIPSO track (?), and the microwaved-based melt-free area seems more extended, so all dataset seem more in phase.
  We very much agree with this comment, and that the manuscript would benefit from better integration of the satellite-based melt results with the other results (e.g., AWS

[Figure]
 Danish Meteorological Institute

temperatures, CALISPO). To achieve this, we have made the following modifications to the manuscript.

Firstly, section 3 (surface melting) has been revised to state: *"The AWS-measured near-surface air temperatures from the 13th to 18th of January (Fig. 4) are consistent with the satellite-based melt patterns (Fig. 3). For example, Sabrina AWS and Elaine AWS both show temperatures above the -2◦C threshold for melt during this period, consistent with both sites being located in a region where the satellite-based measurements show melting. By contrast, Schwerdtfeger AWS and Marilyn AWS show temperatures that are either around or below this threshold, consistent with both sites being located in the western sector of the RIS that the satellite-based measurements identify as being melt-free during this period. Moreover, the erroneous regions of melt over the western RIS simulated by the firn model during this period are consistent with near-surface air temperatures simulated by HIRHAM5 and MetUM being higher than the temperatures observed by Schwerdtfeger AWS and Marilyn AWS (Fig. 4). In particular, at these two stations the HIRHAM5 near-surface temperatures are consistently above -2◦C from the 16th to 18th."*

Secondly, section 5 (cloud properties) now states: "*Additionally, the occurrence of liquid-based clouds on the 14th and 17th of January in the CALIPSO observations over the eastern and central sectors of the RIS is consistent with the satellite-based measurements showing melting here (Fig. 3).*"

References:

de Roda Husman, S., Lhermitte, S., Bolibar, J., Izeboud, M., Hu, Z., Shukla, S., van der Meer, M., Long, D., and Wouters, B.: A high-resolution record of surface melt on Antarctic ice shelves using multi-source remote sensing data and deep learning, Remote Sensing of Environment, 301, 113 950, https://doi.org/10.1016/j.rse.2023.113950, 2024.

Deb, P., Orr, A., Bromwich, D. H., Nicolas, J. P., Turner, J., and Hosking, J. S.: Summer drivers of atmospheric variability affecting ice shelf thinning in the Amundsen Sea Embayment, West Antarctica, Geophysical Research Letters, 45, 4124–4133, https://doi.org/10.1029/2018GL077092, 2018.

Donat-Magnin, M., Jourdain, N. C., Gallée, H., Amory, C., Kittel, C., Fettweis, X., Wille, J. D., Favier, V., Drira, A., and Agosta, C.: Interannual variability of summer surface mass balance and surface melting in the Amundsen sector, West Antarctica, The Cryosphere, 14, 229–249, https://doi.org/10.5194/tc-14-229-2020, 2020.

Johnson, A., Fahnestock, M., and Hock, R.: Evaluation of passive microwave melt detection methods on Antarctic Peninsula ice shelves using time series of Sentinel-1 SAR, Remote Sensing of Environment, 250, 112 044, https://doi.org/10.1016/j.rse.2020.112044, 2020.

Johnson, A., Hock, R., and Fahnestock, M.: Spatial variability and regional trends of Antarctic ice shelf surface melt duration over 1979–2020 derived from passive microwave data, Journal of Glaciology, 68, 533–546, https://doi.org/10.1017/jog.2021.112, 2022.

[Figure]

Danish Meteorological Institute

Lenaerts, J. T., Ligtenberg, S. R., Medley, B., Van de Berg, W. J., Konrad, H., Nicolas, J. P., Van Wessem, J. M., Trusel, L. D., Mulvaney, R., Tuckwell, R. J., Hogg, A. E., and Thomas, E. R.: Climate and surface mass balance of coastal West Antarctica resolved by regional climate modelling, Annals of Glaciology, 59, 29–41, https://doi.org/10.1017/aog.2017.42, 2017.

Mousavi, M., Colliander, A., Miller, J. Z., and Kimball, J. S.: A novel approach to map the intensity of surface melting on the Antarctica ice sheet using SMAP L-band microwave radiometry, IEEE Journal of Selected Topics in Applied Earth Observations and Remote Sensing, 15, 1724–1743, https://doi.org/10.1109/JSTARS.2022.3147430, 2022

[Figure]

Figure 1: 1mm per day threshold, Maps of West Antarctica showing the daily melt area from the 13th to 18th (from left to right) of January 2016 from (top row; a-f) satellite passive microwave measurements, (middle row; g-l) the offline coupled firn model forced by HIRHAM5 output, and (bottom row; m-r) the offline coupled firn model forced by MetUM output. Melt areas are indicated by the dark shading, while melt-free regions are shown as white.

Hansen et al 2023 doi.org/10.5194/tc-2023-145

Danish Meteorological Institute

[Figure]

Figure 2: 3 mm per day threshold (as in the paper), Maps of West Antarctica showing the daily melt area from the 13th to 18th (from left to right) of January 2016 from (top row; a-f) satellite passive microwave measurements, (middle row; g-l) the offline coupled firn model forced by HIRHAM5 output, and (bottom row; m-r) the offline coupled firn model forced by MetUM output. Melt areas are indicated by the dark shading, while melt-free regions are shown as white.

[Figure]

Figure 3: 5 mm per day threshold, Maps of West Antarctica showing the daily melt area from the 13th to 18th (from left to right) of January 2016 from (top row; a-f) satellite passive microwave measurements, (middle row; g-l) the offline coupled firn model forced by HIRHAM5 output, and (bottom row; m-r) the offline coupled firn model forced by MetUM output. Melt areas are indicated by the dark shading, while melt-free regions are shown as white.

---

## Author Response (AR3)

Hansen et al 2023 doi.org/10.5194/tc-2023-145

[Figure]
 Danish Meteorological Institute

Reply to Editor comments on

**"The importance of cloud properties when assessing surface melting in an offline coupled firn model over Ross Ice shelf, West Antarctica"**

by

Nicolaj Hansen, Andrew Orr, Xun Zou, Fredrik Boberg, Thomas J. Bracegirdle, Ella Gilbert, Peter L. Langen, Matthew A. Lazzara, Ruth Mottram, Tony Phillips, Ruth Price, Sebastian B. Simonsen, and Stuart Webster

Dear Editor,

We are very glad that the reviewers have accepted our manuscript. Thank you for these final comments. We very much appreciate the expertise and time that has been spent on this task and sincerely believe that your efforts have resulted in a considerably improved manuscript. We have implemented the vast majority of your suggested changes/comments and very much hope that you now find the manuscript acceptable for publication. In the few occasions that we did not implement your suggested changes to the text it was because we thought that the original text was clear. In the following, we provide a point-by-point reply to each of these.

Yours sincerely,

Nicolaj Hansen

**Technical corrections:**

- All Figures are quite pixelated, please upload higher resolution images. For instance, in Fig. 2b the low HIRHAM5 of melt days values are hardly visible (see L169-170). In Fig. 4, AWS labels are not readable, the same holds for the legend in Fig. 5.
  We have remade Figures 2, 3, 4, 5, 6, 7, and 8 in a higher resolution, and made the labels larger in Fig 4 and the legends larger in Fig 5. Regarding Fig 2b, the number of melt days is so low that a higher resolution does not make a difference.

Hansen et al 2023 doi.org/10.5194/tc-2023-145

[Figure]

Danish Meteorological Institute

- You could improve the referencing of figures across the manuscript, i.e., by referring to specific sub-panels (e.g., Fig. 2a) where appropriate. You can find a few examples below: L170: "central sector (Fig. 2b)." L171: "with the observations (Fig. 2c)." L173: "entire RIS (Figs. 2d,e)." L183: "15th of January (Fig. 3X)" L184: "18th of January (Fig. 3X)" L187: "Fig. 3X" L190: "this period (Figs. 4a,b)" L191: "this threshold (Fig. 4c,d)" L195: "(Fig. 4c,d)" L254: "(>80%) (Fig. 9)." L258: "to the models (Fig. 9)." L265: "(Fig. 10a)" L267: "(Fig. 9c)" L268: "(Fig. 10b)" L280: "(Fig. 10b)" L284: "(Fig. 9f)"
  We have added these examples and others to the text, which have improved the clarity of the results section.

- L215: What do you mean by "transition in SEB" please, clarify.
  We mean from negative to positive, and we have added: "*This suggests that the primary energy source responsible for the transition from negative to positive SEB is from radiative fluxes and not sensible or latent heat fluxes*"

- L294: By "resolution" do you mean spatial/temporal resolution or both, please clarify.
  We mean spatial resolution, which we have clarified in the text.

- L302: "(two order of magnitude)" larger than what? Please clarify.
  We have now clarified this by adding the following: "*are extremely large (MetUM estimates are two orders of magnitude lower than CERES), which suggests that the more likely reason for this is that the MetUM…*"

**Figures corrections:**

- Caption Fig. 1 L2-4: Remove "and over left … on the map." and "and is over the right … of the map".
  We have removed this.

- Figure 3: To clarify, you could add a label in front of each row e.g., "Passive microwave", "HIRHAM5" and "MetUM".
  We have added OBS, HIRHAM5 and MetUM to the rows, which is consistent with the labeling used in Fig. 2.

- Caption of Fig. 8 L3: Remove "Downward fluxes are positive".
  We have kept this in the caption, as it was also included in Figures 6 and 7. .

- Figure 9: The colour scale could be counterintuitive, as one may expect grey shades for cloudy conditions. You could consider reversing the colour scale.
  We have not implemented this suggestion as its standard practice to plot cloud cover from MODIS as white. See for example, Fig. 2 of Pincus et al. (2023; https://essd.copernicus.org/articles/15/2483/2023/). Also, please refer to the following NASA / MODIS website: https://earthobservatory.nasa.gov/global-maps/MODAL2_M_CLD_FR

- Figure 10 caption L2-3: "The light blue bars… include the parts of the satellite ground track that overlap with the RIS region highlighted in Fig. 1."
  We have changed the caption

[Figure]

Danish Meteorological Institute

Thanks for all the stylistic suggestions, we have implemented almost all of them.

**Stylistic suggestions:**

- L11: "a transition from clear-sky to cloudy conditions, with clouds containing both…"
  Done

- L18: "This can lead to ice shelves thinning…"
  Done

- L20-21: "and thus enhanced global sea-level rise…"
  Done

- L22: "ice/snow at 0ºC, …"
  Done

- L25: "ice shelf melting are usually driven/triggered by local…"
  Done

- L31: "grid box of ~10 km resolution or less."
  Done

- L32: "enhance the representation of crucial…"
  Done

- L33: "coastal margins, and better resolve small ice shelves at spatial scales of…"
  Done

- L36: "changes of snow/firn properties in the upper…"
  Done

- L38: "feedback, and meltwater retention and refreezing in the firn layer"
  Done

- L40: "these effects vary considerably; the spin-up time selected to simulate the evolution of the snow/firn layer also affects model performance."
  We have taken your suggestion partly into account, and revised the sentence to be: 'The ability and sophistication of land surface and subsurface snow schemes in regional atmospheric models to represent these effects varies considerably, with the choice of spin-up time selected to simulate the evolution of the snow/firn layer also affecting the model performance'

- L44-45: "atmospheric models. The representation of cloud properties,…"
  We have altered the first line of this paragraph to be consistent with the previous paragraph, and also follow your suggestion. It now states 'The representation of cloud properties, particularly cloud phase and microphysics, are also a major challenge for regional atmospheric models.'

- L60-61: "(2017), that represents … feedback, …"
  Done

- L64: "2023). In particular, the January 2016 event is attributed to…"
  Done

- L67: "models that require further improvements."
  Done

- L69: "Trusel et al. (2015) suggest…"
  Done

- L70-71: "under a high-emission climate scenario (RCP or SSPX-X.X)" Please mention the scenario used.
  Done

[Figure]

- L73-75: "Thus, improving the representation of surface melting (and hence surface mass balance) that is an indicator for … 2019) is essential … stability, and estimate/quantify its contribution…"
  Done
- L85: "dry fresh snow at temperatures below -5ºC"
  Done
- L127: "temperature exceeds 0ºC"
  Done
- L132: "and nighttime, and clear and cloudy conditions"
  Done
- L140: "see Fig. 1 for AWS locations"
  Done
- L141: "occur, as suggested"
  Done
- L151: Do you mean "depolarised" by "depolarising"?
  This was unclear.  The sentence has been modified to 'These observations use differences in the polarization properties of light backscattered from non-spherical ice particles and spherical water droplets to determine important information on the cloud phase (Hu et al., 2009).'

- L165: "showed that this"
  Done
- L189: "For example, Elaine and Sabrina AWS both …"
  Done
- L192-193: "of the RIS, where the satellite … identify melt-free conditions during…"
  Done
- L200: "compares timeseries of surface radiative fluxes for this period that are spatially…"
  Done
- L201: "From the 13th to 14th, … during nighttime ranging from -20 to -40 W m-2".
  Done
- L203: "freezing conditions, …"
  This suggestion was not implemented as the grammar was correct.
- L206: "of freezing, "
  Done
- L210: "being more extensive over the western"
  Done
- L211: "as well as with HIRHAM5 … over this region than MetUM"
  Done
- L212-214: "cycle of modelled SEB broadly follows… minimum modelled SEB shows negative… values for [provide days] and zero/positive values for [provide days]"
  Done
- L217: "radiative fluxes contribution to surface melt is balanced."
  This suggestion has not been implemented.
- L220: Remove "(as expected)"
  Done
- L231: Remove "here"
  Done

[Figure]
 Danish Meteorological Institute

- L266: "January, the MODIS imagery"
  Done
- L268: "over the same region where models show"
  Done
- L271: Remove "here"
  Done
- L279: "1 kg m-2, i.e., two orders"
  Done
- L292: "observations suggesting that both…"
  Done
- L296: "which results in significant … that are nearly non-depolarised"
  Done
- L299: "for the 16th to the 18th of January"
  Done
- L302-303: "which suggests that discrepancies result from MetUM severely underestimating … water, and not from uncertainties in the observations."
  Done
- L306: "case, the associated additional surface melting would … measurements, showing …"
  Done
- L309-310: "For example, issues in simulating cloud microphysics may affect other …"
  Done
- L312-313: "Unfortunately, neither AWS … output fields are available…"
  Done
- L314-315: "will require an increasing number of… able to comprehensively measure radiation and cloud properties"
  Done
- L316-320: "surface observation network compiling full … require maintenance to ensure sufficient measurements quality for use in future studies. A similar study focusing on melt events that include such measurements… worthwhile. By repeating the … scheme, we could examine whether model upgrades could improve the representation of cloud microphysical properties (… 2023)."
  Done
- L325: "particularly since the model simulates Antarctic …"
  This has not been implemented. The grammar was fine in the text we had originally used.
- L330: "higher resolution grids,…"
  Done
- L353: "This results from models simulating… "
  Done

---

## Author Response (AR4)

Hansen et al 2023 doi.org/10.5194/tc-2023-145

[Figure]

Reply to Editor comments on

**"The importance of cloud properties when assessing surface melting in an offline coupled firn model over Ross Ice shelf, West Antarctica"**

by

Nicolaj Hansen, Andrew Orr, Xun Zou, Fredrik Boberg, Thomas J. Bracegirdle, Ella Gilbert, Peter L. Langen, Matthew A. Lazzara, Ruth Mottram, Tony Phillips, Ruth Price, Sebastian B. Simonsen, and Stuart Webster

Dear Editor,

We are very glad that the reviewers have accepted our manuscript. Thank you for these final comments. We very much appreciate the expertise and time that has been spent on this task and sincerely believe that your efforts have resulted in a considerably improved manuscript. We have implemented the vast majority of your suggested changes/comments and very much hope that you now find the manuscript acceptable for publication. In the few occasions that we did not implement your suggested changes to the text it was because we thought that the original text was clear. In the following, we provide a point-by-point reply to each of these.

Yours sincerely,

Nicolaj Hansen

**Technical corrections:**

- L24: The authors could choose between "driven" and "triggered", as most appropriate.
  We have chosen "driven"
- L75: The authors could choose between "estimate" and "quantify", as most appropriate.
  We have chosen "estimate"